# Spatial arrangement of several flagellins within bacterial flagella improves motility in different environments

Marco J. Kühn[1], Felix K. Schmidt[2], Nicola E. Farthing[3,4], Florian M. Rossmann[1], Bina Helm[1], Laurence G. Wilson [3], Bruno Eckhardt[2] & Kai M. Thormann[1]

Bacterial flagella are helical proteinaceous fibers, composed of the protein flagellin, that confer motility to many bacterial species. The genomes of about half of all flagellated species include more than one flagellin gene, for reasons mostly unknown. Here we show that two flagellins (FlaA and FlaB) are spatially arranged in the polar flagellum of *Shewanella putrefaciens*, with FlaA being more abundant close to the motor and FlaB in the remainder of the flagellar filament. Observations of swimming trajectories and numerical simulations demonstrate that this segmentation improves motility in a range of environmental conditions, compared to mutants with single-flagellin filaments. In particular, it facilitates screw-like motility, which enhances cellular spreading through obstructed environments. Similar mechanisms may apply to other bacterial species and may explain the maintenance of multiple flagellins to form the flagellar filament.

[1] Institut für Mikrobiologie und Molekularbiologie, Justus-Liebig-Universität Gießen, 35392 Gießen, Germany. [2] Fachbereich Physik und LOEWE Zentrum für Synthetische Mikrobiologie, Philipps-Universität Marburg, 35032 Marburg, Germany. [3] Department of Physics, University of York, Heslington, York YO10 5DD, UK. [4] Department of Mathematics, University of York, Heslington, York YO10 5DD, UK. These authors contributed equally: Marco J. Kühn, Felix K. Schmidt, Nicola E. Farthing. Correspondence and requests for materials should be addressed to L.G.W. (email: laurence.wilson@york.ac.uk) or to B.E. (email: bruno.eckhardt@physik.uni-marburg.de) or to K.M.T. (email: kai.thormann@mikro.bio.uni-giessen.de)

Active motility enables bacteria to establish themselves in their favorable environmental niche. For propulsion many bacterial species employ flagella, long helical proteinaceous filaments extending from the cell surface into the surrounding environment. They are rotated at the filaments' base by membrane-embedded motors, which commonly allow rotational switches to enable navigation[1]. Each flagellar filament is composed of thousands of copies of flagellin subunits[2,3] in a process that is tightly regulated at the expression level. Typically, expression of the flagellin genes requires alternative sigma factors, and production and export of the flagellins is initiated once assembly of the basal body and the hook joint structure is completed[4]. The flagellins are then transported through the filament and assembled at the tip of the growing flagellum[5–7].

Many bacterial species, such as Escherichia coli, harbor a single flagellin gene so that the resulting flagellar filament consists of only one type of subunit. However, about 45% of the flagellated bacterial species possess more than a single flagellin gene[8], with numbers ranging up to fifteen copies as reported for Magnetococcus sp. MC-1[9]. Salmonella Typhimurium with two distinct flagellins undergoes phase variation and only utilizes one of the two flagellins at a time to build the filament[10]. In contrast, most other bacterial species with multiple flagellins, such as Bacillus thuringiensis, Bdellovibrio bacteriovorus, Campylobacter jejuni, Caulobacter crescentus, Helicobacter pylori, Sinorhizobium meliloti, and Vibrio spp., have been shown to assemble the flagellar filament from all or at least most flagellins encoded in their genome[8,11–20]. For B. bacteriovorus, C. crescentus, and H. pylori, it was demonstrated that the different flagellins exhibit a spatial arrangement within the filament[12,14,16].

The role of different flagellins within the flagellar filament is still mostly unclear. While in some systems one flagellin exhibits the major structural or functional role, a large degree of functional redundancy occurs in other multi-flagellin systems. However, depending on the species, loss of certain flagellins in such systems may result in changes in the filament's morphology and function, e.g., with respect to swimming speed, adhesion, or cytotoxicity[8,12,20,21]. It has thus been suggested that the composition of the filament may be influenced by the environmental conditions[8,18].

Species of the genus Shewanella possess a single polar flagellum in the form of a left-handed helix, which pushes the cell during counterclockwise (CCW) rotation and pulls the cell upon switching to clockwise (CW) rotation. With few exceptions, the gene cluster encoding the components of the Shewanella polar flagellar system harbors two distinct flagellins[22–24]. In this study, we explored the role of the two flagellins, FlaA and FlaB, of the polar flagellar system in S. putrefaciens CN-32. We determined how FlaA and FlaB production may result in spatial arrangement within the flagellar filament, which enabled us to create mutants with defined filament compositions. Subsequent microscopy and three-dimensional tracking of swimming trajectories demonstrated that the composition of the filament significantly affects the flagellar morphology and the motion during free-swimming and spreading through soft agar. In agreement with corresponding simulations on flagellar stability under rotation, the results strongly implied that the basal flagellar part formed by the flagellin FlaA benefits swimming under a range of different conditions. With the switch to FlaB in the remainder of the flagellin, the cells maintain the ability to wrap the filament around the cell body and execute a screw-like movement[23], which is highly beneficial for spreading through complex environments.

## Results

### Characterization of S. putrefaciens FlaA and FlaB. 
S. putrefaciens possesses a monopolar and a lateral flagellar system. We previously showed that the polar system predominantly mediates propulsion, chemotactic behavior and backward screwing motion[22,23,25]. In the following, only this polar filament will be addressed, and all experiments described here were conducted using a strain unable to produce the lateral flagellar filaments due to the lack of the corresponding flagellin-encoding genes ($\Delta flaA_2B_2$). For simplification, this strain will henceforth be referred to as wild type (FlaAB).

The two genes encoding the polar flagellins, flaA (Sputcn32_2586) and flaB (Sputcn32_2585), are arranged next to each other[22] and encode proteins of 273 (FlaA) and 271 (FlaB) amino acids with estimated molecular masses of 28.6 (FlaA) and 28.4 (FlaB) kDa. They exhibit a high degree of conservation (86% identity; 91% similarity); most differences occur within the flagellin variable region which presumably faces the environment (Supplementary Fig. 1)[3,6].

We previously showed that both FlaA and FlaB are produced at estimated molecular masses of ~37 and ~35 kDa, respectively. This is higher than the masses deduced from the amino acid sequence (~28.5 kDa)[23], indicating modification of both flagellins, e.g., by glycosylation. To determine if the flagellins of S. putrefaciens are glycosylated, we introduced in-frame deletions into the genes encoding PseG (Sputcn32_2626) and Maf-1 (Sputcn32_2630), as the deletion of the orthologs in S. oneidensis results in completely non-glycosylated flagellins[26,27]. Immunoblotting analysis of PAGE-separated crude extracts (Supplementary Fig. 2a) revealed that in both $\Delta pseG$ and $\Delta maf$-1 mutants the flagellins occur as a single band at a position according to their estimated molecular mass. Thus, within the cell population, both FlaA and FlaB are produced and both are likely modified by glycosylation. The difference in the molecular mass, presumably caused by a more extensive modification of FlaA, enabled us to determine the presence of each flagellin by immunoblotting, which helped to unravel the regulation of flagellin production later in this study.

### Spatial arrangement of FlaA and FlaB in the filament. 
To explore the distribution of FlaA and FlaB in filament assembly at the single cell level, we performed fluorescence microscopy on cells in which FlaA, FlaB, or both, can be selectively labeled by coupling with a maleimide-ligated fluorophore (FlaA-Cys and FlaB-Cys). The corresponding serine/threonine to cysteine substitutions in the flagellin proteins do not affect swimming motility or flagellin decoration by glycosylation (Supplementary Fig. 3 and ref.[23]). We found that FlaA and FlaB exhibit a spatial arrangement within the flagellar filament (Fig. 1a, e; Supplementary Fig. 4). FlaA almost exclusively forms the base of the filament close to the cell body, which accounts for about 8–28% (17% average) of the fully assembled flagellum. The residual filament is predominantly composed of FlaB with some interspersed FlaA. Mutants completely lacking FlaB ($\Delta flaB$, FlaA stub) produced very short filaments, which barely supported swimming (Supplementary Fig. 5a, e, i, m) while $\Delta flaA$ mutants (FlaB-only) formed flagellar filaments that were not noticeably different from the wild-type filament in morphology or length (Fig. 1b, f; Supplementary Fig. 6). This spatial flagellin arrangement strongly suggested sequential production and/or export of FlaA and FlaB.

Previous studies on the closely related species S. oneidensis indicated that flaA and flaB are not organized in an operon with a single promoter but are regulated individually[24,28]. Accordingly, q-RT-PCR revealed that in S. putrefaciens the flaB transcript is about twice as abundant as that of flaA (Supplementary Fig. 7a), also indicating individual regulation. We therefore monitored the production of FlaA and FlaB in mutants deleted in genes rpoN, flrA, and fliA, encoding main regulators of flagellation. Generally, in the group of polarly flagellated gammaproteobacteria, RpoN ($\sigma^{54}$) acts as a major general regulator of flagellation, FlrA is required for production of numerous structural flagellar components, and the sigma factor FliA ($\sigma^{28}$) is required for the

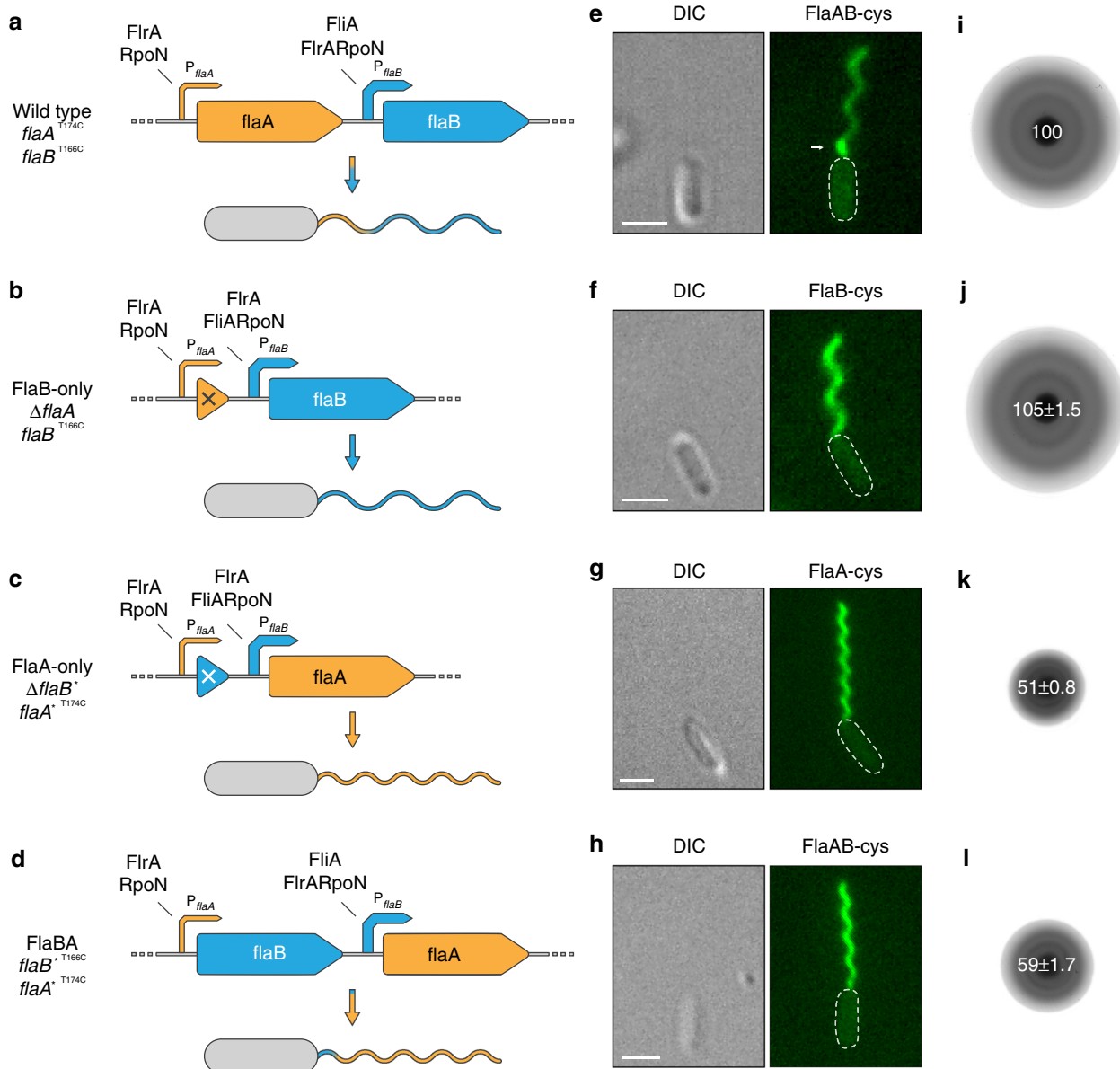

**Fig. 1** Overview of the different types of polar flagellar filaments constructed in this study. In all strains the genes for the lateral flagellins were deleted. **a–d** Genetic organization of the polar flagellins and genetic modifications to obtain different filament types. In the wild-type context *flaA* lies upstream of *flaB* and transcription is controlled by a FliA-independent promoter (cp. Supplementary Fig. 2b). In contrast, transcription of *flaB* is controlled by a stronger, FliA-dependent promoter and mediates production of the major part of the flagellar filament. Expression of both genes relies on RpoN and FlrA. A functional filament is only produced when at least one of the flagellar genes is expressed from the *flaB* promoter. If no flagellin is expressed from the *flaB* promoter only short filament stubs are formed (see Supplementary Fig. 5). Gene deletions are marked with a cross, swapping of the flagellin gene sequences is marked with an asterisk. P*flaA/B* = *flaA/B* promoter. **e–h** Micrographs of cells with fluorescently labeled flagellar filaments displaying the outcome of the genetic editing of the flagellin genes. Panel **e** also shows the spatial distribution of the flagellins FlaA and FlaB in the wild-type filament (see also Supplementary Fig. 4). The FlaA portion is marked with an arrow. The increased fluorescence of FlaA compared to that of FlaB is likely due to a better accessibility of the cysteine residue during the fluorescent labeling process. Scale bars represent 2 μm. **i–l** Radial expansion of cells with different types of flagellar filaments in 0.25% soft agar. Wild-type and FlaB-only cells spread well in this structured environment while the spreading ability of FlaA-only and FlaBA cells is strongly impaired. The numbers in the center of the colonies indicate the relative spreading compared to the wild type in percent (% ± s.d.) of three individual experiments

production of late flagellar components such as flagellin. FliA is released from its inactivating anti-sigma factor FlgM upon completion of the basal body and the hook to avoid premature production of large amounts of flagellin[4]. Western immunoblotting revealed that, in *S. putrefaciens*, production of both FlaA and FlaB strictly requires RpoN and FlrA. In contrast, in a *ΔfliA* mutant FlaA as well as the flagellin chaperone FliS, which is

required for export, were produced in normal amounts while FlaB was absent (Supplementary Fig. 2b, c). Hence, production of FlaA can already start before the hook-basal body complex is finished, but FlaB production requires hook completion and the subsequent export of the anti-sigma factor FlgM. This regulatory arrangement indicated that a pool of FlaA is already present and ready to be exported and assembled into the filament as soon as

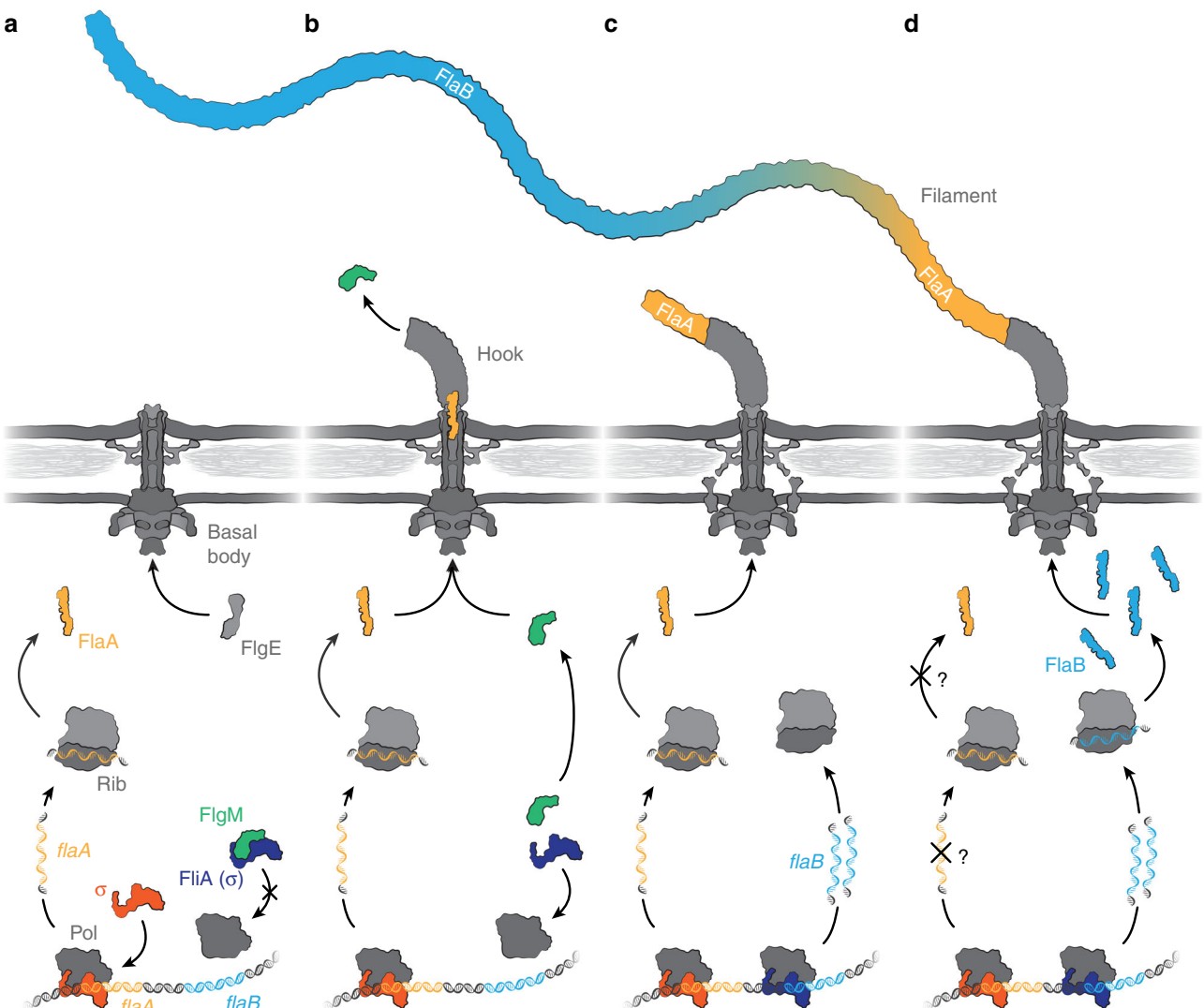

**Fig. 2** Schematic model of the sequential production and export of FlaA and FlaB. The figure summarizes our current working model of how spatial arrangement of the two flagellins is achieved in *Shewanella putrefaciens*, depicted in four subsequent stages of the flagellar assembly. **a** The current model of flagellin expression suggests that transcription is inhibited as long as the alternative sigma factor 28 (FliA) is blocked by the anti-sigma factor FlgM. In *S. putrefaciens*, this only applies to FlaB while FlaA production does not depend on FliA (see Supplementary Fig. 2b). Therefore, a pool of FlaA is already present while the basal body is still being assembled. **b** Once the hook-basal-body complex is completed the export apparatus changes its specificity from hook proteins (FlgE) to the flagellins and other late assembly proteins. At this stage also the anti-sigma factor FlgM gets exported and FliA-dependent transcription of FlaB is initiated. **c** The already produced FlaA monomers get exported first and assemble at the base of the flagellar filament. **d** The increasing production of FlaB monomers passes that of FlaA and potentially FlaA transcription and/or translation may be reduced or even completely terminated (indicated by the question marks). Thus, FlaB constitutes the majority residual part of the flagellar filament. (Pol, RNA polymerase; Rib, ribosome; σ, sigma factor)

the hook is completed, while FliA-dependent production of FlaB is initiated (Fig. 2).

Accordingly, a strain with only *flaB* under control of the *flaA* promoter produced very short FlaB filaments (Supplementary Fig. 5b, f, m). In contrast, placing only *flaA* under control of the *flaB* promoter resulted in flagellar filaments exclusively consisting of FlaA (Fig. 1c, g), which were of almost normal length (~6.5 μm) and also exhibited a left-handed helix. However, pitch, diameter and therefore number of turns of FlaA-only helical filaments differed significantly from FlaB-only or wild-type filaments (Supplementary Fig. 6). Finally, when placing *flaB* under the control of the *flaA* promoter and vice versa, a flagellar filament with FlaB forming a short basal segment was formed, while the residual filament consisted of FlaA (Fig. 1d, h;

Supplementary Fig. 4—FlaBA). Accordingly, the overall filament of these FlaBA mutants had a highly similar geometry as the FlaA-only filament (Supplementary Fig. 6).

Based on these results we propose that the sequential production pattern depends on two individual promoters driving the expression of *flaA* and *flaB*, respectively. The length of the proximal filament segments formed under control of the weaker FliA-independent *flaA* promoter showed a wide range, indicating flagellin production from this promoter varies substantially at the single cell level. Accordingly, overproduction of FlaA from a plasmid resulted in aberrantly long flagellar filaments (Supplementary Fig. 7b, c), indicating that the amount of available flagellin monomers is an important length determining factor. However, the data did not explain why flagellin production

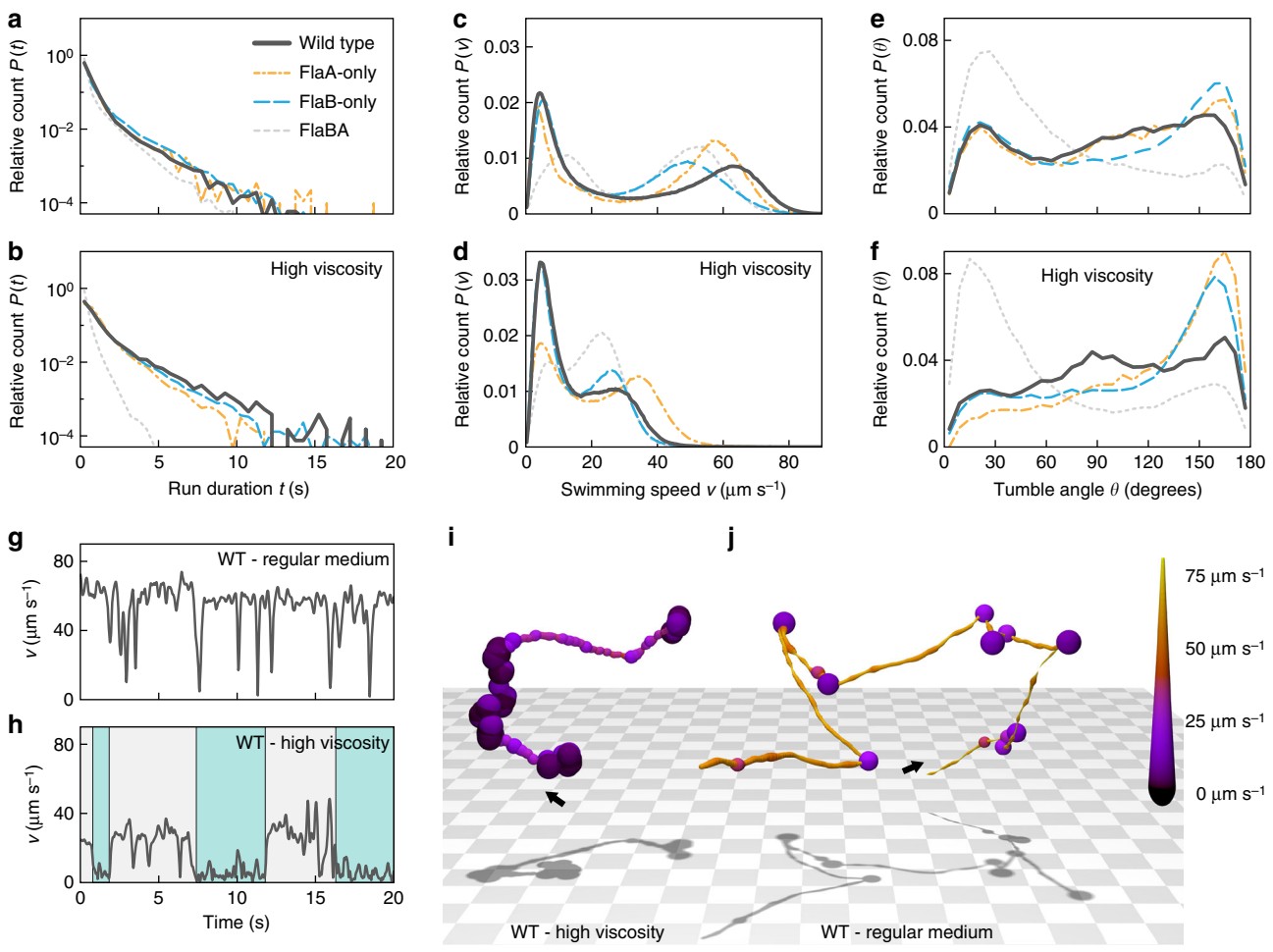

**Fig. 3** Free-swimming behavior mediated by the four different flagellar filament types. Several thousand cell tracks were obtained for each strain by holographic microscopy. The diagrams show the relative counts to account for different sample sizes. **a, b** Run duration histograms for wild-type (FlaAB, dark grey solid lines), FlaA-only (orange mixed dashed lines), FlaB-only (blue long dashed lines) and FlaBA (light grey short dashed lines) cells in regular medium (upper panel) and medium with increased viscosity (lower panel). $P(t)$ indicates the probability of observing a run with a duration in the range $t$ to $t+dt$. **c, d** Velocity distribution for the different filament types in regular medium and medium with increased viscosity, respectively. $P(v)$ indicates the probability of observing a cell swimming with a speed in the range $v$ to $v+dv$. **e, f** Turning angle histograms for the different filament types in regular medium and medium with increased viscosity, respectively. Low and high turning angles correspond to weak deviations from straight swimming and near-reversals, respectively. $P(\theta)$ indicates the probability that a particular re-orientation event results in a change of direction in the range $\theta$ to $\theta+d\theta$. **g, h** Velocity series for a single wild-type cell in regular medium and a similar cell in high viscosity medium, respectively. Under conditions of increased viscosity the cell exhibits longer periods of swimming at a slower speed (the cyan-colored areas in panel **h**), indicative of screw formation. **i, j** Stereotypical cell tracks for wild-type cells in medium with increased viscosity (left) and regular medium (right). The tracks correspond to the velocity series in panels **g** and **h**. The starting points of the tracks are marked by arrows. The color coding and the line thickness represent the cells' swimming speed. The corresponding movie is provided as Supplementary Movie 1

from the *flaA* promoter only produces filament stubs, which indicates cessation of the expression from this promoter that may occur at the onset of *flaB* expression.

**Filament composition affects flagella-mediated swimming.** The different morphologies of the FlaA- and FlaB-only filaments suggested that the filament composition affects the swimming behavior of the cells[8,29]. We therefore compared the spreading ability of the wild-type cells (FlaAB), FlaA-only (with *flaA* being expressed from the *flaB* promoter, resulting in filaments of normal length), FlaB-only and FlaBA mutants through soft agar. Here, the cells have to navigate through an intricate network of polysaccharide strands (Fig. 1i–l). Under these conditions, FlaB-only mutants spread as well as wild-type cells, while cells driven by FlaA-only or FlaBA filaments exhibited a pronounced decrease in spreading. As the inability of these filaments to

promote normal spreading in soft agar could be due to general differences in free swimming, we compared the free-swimming patterns of wild-type (FlaAB) cells to those driven by FlaA-only, FlaB-only and FlaBA filaments by three-dimensional holographic tracking (Fig. 3; example tracks given in panels i and j). This was carried out at normal and, to simulate flagellar performance at high load, at elevated viscosity (10% wt·vol⁻¹ Ficoll 400). From the trajectories obtained, three major swimming parameters were deduced: run duration, swimming velocity, and turning angle.

The run duration represents the time interval of swimming between directional changes. A shift toward shorter run durations may negatively affect effective spreading in soft agar while very long runs may hamper navigation. We found that under conditions of normal viscosity the distribution of run durations was very similar for all four strains with a descending slope

from short runs (1 s or less) toward longer runs up to about 20 s (Fig. 3a). Under conditions of elevated viscosity (Fig. 3b), the distributions of run durations remained similar except for the FlaBA mutant that exhibited a pronounced shift towards shorter runs, indicating a lower spreading ability of this strain.

The second major parameter, the swimming velocity, directly affects the speed of spreading. At normal viscosity, all four strains displayed a similar top speed and velocity distributions with two major subpopulations, a slow and a fast one (Fig. 3c). The slow population, which did not include completely non-motile cells, was comparable in size for the wild type and the FlaA-only and FlaB-only strains, but much broader for the FlaBA mutant. The mean speed increased from 5 µm s$^{-1}$ for the first three strains to 15 µm s$^{-1}$ for the FlaBA mutant. The velocity of the fast-moving subpopulation was different for all four strains: The wild-type cells had the highest speed (~63 µm s$^{-1}$), followed by FlaA-only cells (~57 µm s$^{-1}$), FlaBA cells (~53 µm s$^{-1}$) and FlaB-only cells (~49 µm s$^{-1}$). At elevated viscosity (Fig. 3d), the velocity distribution with a slow and a fast swimming major population persisted for all four strains, albeit at a lower overall speed. Under these conditions, FlaA-only cells were the fastest (~38 µm s$^{-1}$), confirming earlier reports stating that a smaller helix diameter of a bacterial flagellum promotes swimming at elevated viscosity[29]. The FlaA-only mutant was followed by wild-type and FlaB-only cells (~28–30 µm s$^{-1}$) and FlaBA (~22 µm s$^{-1}$). Notably, the size of the slow subpopulation of wild-type and FlaB-only cells increased compared to normal viscosity, but not the one of the other two strains.

The third major parameter, the turning angle, indicates how efficiently the cells can switch direction. Very high angles between 150° and 180° would indicate forward-backward (run-reverse) swimming without efficient directional changes and thus a decrease in spreading efficiency. All four strains showed a wide variation of turning angles from 0° to 180°. Under conditions of normal viscosity (Fig. 3e), the turning-angle profiles of wild-type, FlaA-only and FlaB-only cells were rather similar. However, at elevated viscosity (Fig. 3f), the wild-type cells maintained the wide distribution of turning angles, while for FlaA-only and FlaB-only mutants a large subpopulation exhibited directional changes at very high angles in a run-reverse mode, which would decrease efficient spreading. The FlaBA-mutant cells showed an aberrant behavior under both conditions, as a major subpopulation of the cells displayed rather low turning angles (0–60°), resulting in a higher directional persistence.

The trajectory data on free-swimming cells demonstrated that the composition of the flagellar filament significantly affects the free-swimming behavior of the cells. However, the free-swimming capability of the different strains did not correlate well to the spreading capability in soft agar. This was particularly eminent in the case of cells driven by FlaA-only filaments, which did not spread well in soft agar but outperformed FlaB-only mutants, the best spreaders in soft agar, in almost every aspect of free swimming. Discrepancies with the agar-based experiments are to be expected, as the tracking experiments measure the individual characteristics of cells in the absence of external perturbations, such as mechanical interactions with their environment. In contrast, the agar assays capture a holistic picture of motility, integrating speed, reversal rate, response to mechanical environment, etc. across a whole population[30], which cannot be adequately simulated by simply increasing the viscosity as in our free-swimming assays. We concluded that the poor spreading capability of the FlaA-only and FlaBA mutants in structured environments is not due to obvious deficits in free-swimming capabilities. The source of the differences must therefore be connected with other properties of the flagellum.

**The FlaA segment stabilizes the flagellar filament**. The analysis of swimming speed at high viscosity showed a pronounced increase in size of the slow subpopulation of wild-type and FlaB-only cells, but not of FlaBA and FlaA-only strains (Fig. 3c, d). Corresponding single-cell trajectories indicated that a drop in speed frequently followed a directional switch after which the cells continued at lower speed, but accelerated again after another switch of direction (Fig. 3h). We previously showed that under conditions of increased viscosity and while swimming backward with the filament pulling the cell, *S. putrefaciens* can exhibit a filament instability, which results in wrapping of the flagellar filament around the cell and a slow movement in a screw-like fashion[23]. Therefore, we anticipated that the drop in swimming speed of wild-type and FlaB-only cells was due to screw formation. An analysis of the four strains' flagellar behavior at normal and high viscosity (Fig. 4a) confirmed that at normal viscosity only few cells of the wild-type population (<5%) exhibited screw formation during backward swimming[23]. Notably, under these conditions, already about half of the cells of FlaB-only mutants displayed screw formation, while this flagellar behavior was completely absent for FlaA-only or FlaBA cells. The screw-like swimming phenotype became significantly more pronounced at high viscosity, where about half of all backward-swimming wild-type and more than 85% of FlaB-only cells displayed the screwing behavior. Notably, also at increased viscosity, not a single FlaA-only or FlaBA mutant cell exhibited screwing motility, strongly indicating that a filament mainly or exclusively formed by FlaA is more stable when rotating CW and pulling the cells. This finding strongly suggested that the filaments' base formed by FlaA stabilizes the flagellum to prevent premature screw formation during free swimming. To further support this hypothesis, we introduced a second synthetic *flaA* gene into the chromosome directly downstream of the native *flaA* (FlaAAB; Supplementary Fig. 8a, c, e). As anticipated, the higher level of FlaA production increased the length of the basal FlaA segment on average from ~17% to ~24% of the overall filament. In this strain, the number of cells forming the screw at high viscosity dropped from about 50% to about 25%, and at normal viscosity almost no screws (2%) were observed (Supplementary Fig. 9a). Similarly, we constructed a FlaBBA mutant strain to probe if a longer FlaB segment at the filament's base affects the stability of the flagellum (Supplementary Fig. 8b, d, f). The length of the FlaB basal section increased from ~6.8% to ~8.1%, but we did not observe any screw formation. Based on these findings, we conclude that the FlaA segment stabilizes the flagellar filament against screw formation to an extent which depends on the length of this segment.

The findings of the screw formation analysis correlate well with the observed differences in free-swimming speed distributions and spreading through obstructed environments such as soft agar. Wild-type and FlaB-only filaments readily form flagellar screws under conditions of high viscosity, which increases the population of slow free-swimming cells (cp. Fig. 3c, d) as free swimming in screwing mode does not allow effective propulsion[23]. Accordingly, this increase in the slow population of free-swimming cells is absent in FlaBA and FlaA-only mutants, which lack the ability of screw formation. However, the latter two strains are hampered in spreading through structured and obstructed environments, the soft agar plates (Fig. 1k, l), strongly indicating that screw formation is significantly contributing to movement when the cells can make contact with a solid environment. In contrast, the differences in efficient turning angles observed under free-swimming conditions at high viscosity cannot be attributed to screw formation: compared to the wild-type FlaB-only cells, that are prone to screw formation, as well as FlaA-only cells, which are incapable of screw formation, tend to turn at high angles in less efficient forward–backward movement. Notably, in contrast to the mutants, the wild-type cells driven by the native FlaAB filament always performed very well under all conditions tested.

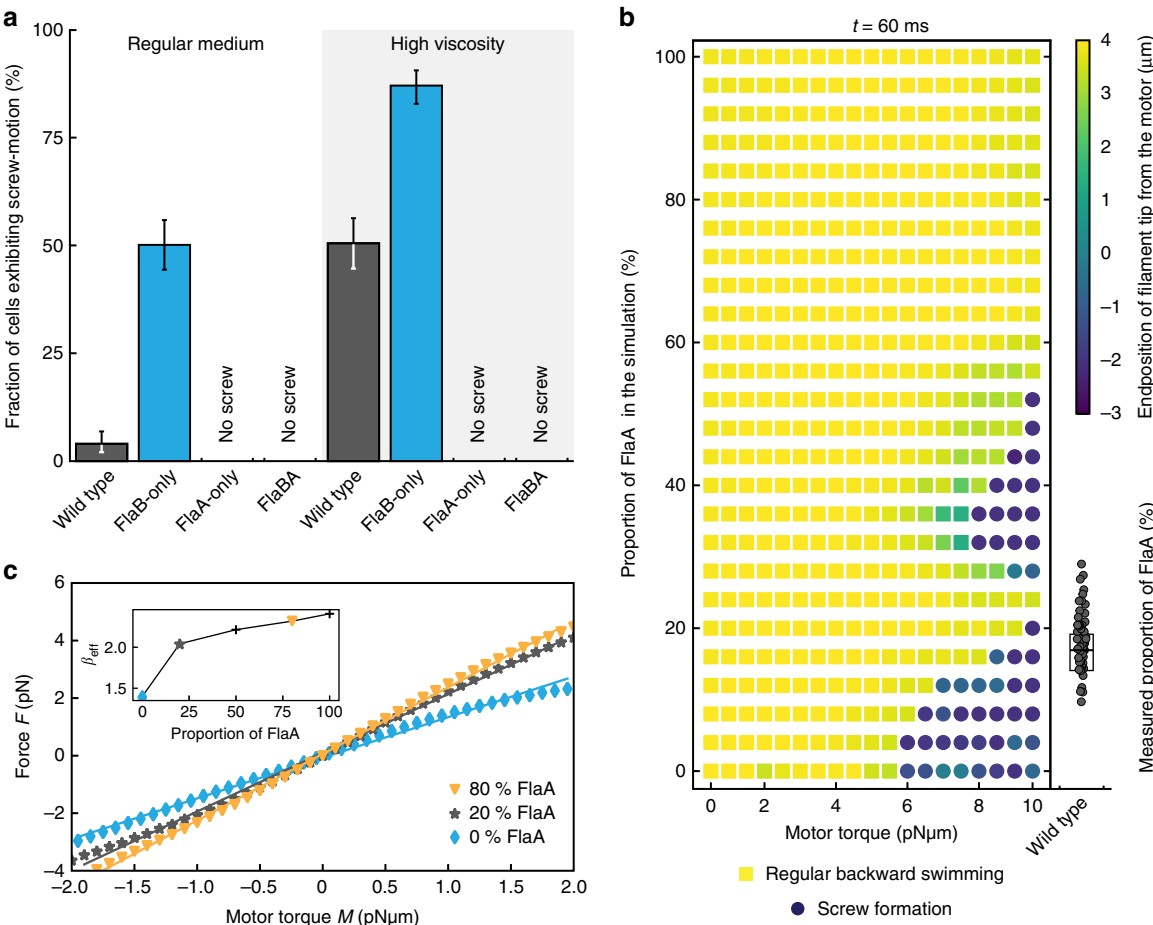

**Fig. 4** Experimentally observed and simulated screw formation. **a** The probability of screw formation of wild-type (FlaAB) and FlaB-only cells increases in high viscosity, while for FlaA-only and FlaBA screw formation was never observed. Significance was tested for all filament combinations under both conditions and for each filament between the two conditions. If no screw was observed at all, significance was not tested. All tested combinations were significant ($P < 0.05$, Bonferroni corrected). Error bars indicate 95% confidence intervals. About 300 cells were counted for each strain. **b** Observation of screw formation for varying flagellin compositions at different motor torques after a simulation time of $t = 60$ ms. The simulation was carried out for flagella with an increasing number of FlaA segments, starting with a flagellum completely composed of FlaB (bottom of the diagram) and successively exchanging the segments to a FlaA configuration starting from the filament's base. Yellow squares indicate regular backward rotation, blue dots indicate screw formation. The color coding represents the z-position of the flagellum's free end, with negative values indicating a position below the motor segment (position 0). For comparison with the actual flagellin composition in the wild type experimentally measured FlaA proportions are given on the right. Data points are displayed as individual values (gray dots) measured for 50 filaments. The box spans the central half of the data, the black bar indicates the median. A movie of the simulation with FlaA-only and FlaB-only filaments is provided as Supplementary Movie 2. **c** Force-torque relation extracted from the numerical simulation. The mean force $F$ on the flagellum varies linearly with motor torque $M$ for both forward and backward swimming. Shown are the relations for a FlaB-only flagellum (blue diamonds), a 20% admixture of FlaA (gray stars) and an 80% admixture (orange triangles). The continuous lines are linear fits to the data. The slope $\beta_{eff}$ in $F = \beta_{eff} M$ versus the FlaA content is shown in the inset: it increases rapidly up to about 20% FlaA and more slowly for larger fractions. A higher value of $\beta_{eff}$ indicates a more efficient transfer of torque into driving force

**Numerical simulations on behavior of the flagellar filament.** The mechanical properties of filaments and their effects on the dynamics can also be studied in numerical simulations. In an established model for filaments, the flagellum is replaced by a string of beads that are connected to each other by elastic bonds with preferred bonding angles, so that in equilibrium a helix with prescribed radius and pitch forms. The interaction with the surrounding fluid is given by resistive force theory (see ref. [23]). Such a model reproduced the screw formation of the helix in dependence on the motor torque and thus the force acting on the flagellar helix. Here, we extend our previous model to the case of two flagellins by adjusting the parameters locally to those of the corresponding flagellins, while keeping all other parameters fixed. Specifically, FlaA forms narrow elongated helices (radius: 0.175 μm; pitch 1.18 μm) whereas FlaB forms wider and shorter helices (radius: 0.315 μm; pitch 1.91 μm; Supplementary Table 6).

In the simulations we can switch between the two flagellins at continuous positions along the flagellum. This allows us to study the properties of the flagellum as a function of the partitioning into FlaA and FlaB and of the motor torque. As the screw forms after about 60 ms[23], the simulation was conducted for this period of time. For the wild-type flagellum composed of FlaA close to the cell and FlaB further away, we find that it is stable to screw formation for motor torques of less than about 5 pNμm. For larger torque, screws form, but the required torque varies with FlaA fraction: It reaches a local maximum in stability for a fraction of about 24%, where a torque of 8.5 pNμm is needed to form the screw (Fig. 4b). This simulated region of enhanced stability corresponds well to our experimental findings that an increase to an average of 24% proximal FlaA, as observed in the FlaAAB mutant, increases the filament's stability towards screw formation compared to that of the wild type (Supplementary Fig. 9c).

The model shows a second region with increased stability for FlaA fractions above about 40%, but that could not be reliably probed with our mutants. Turning the order of FlaA and FlaB around, no screws are formed until the FlaB part exceeds 40% (Supplementary Fig. 9d), and then the required torque falls with increasing fraction. This is in line with our experiments in which FlaB forms up to 10–12% of the flagellar filament's base and no screw formation occurs. Thus, the simulation not only reproduces our experimental findings but also suggests that introducing a proximal FlaA filament segment with a smaller diameter and pitch is already sufficient to result in the observed stabilization of the flagellar filament against screw formation, indicating the geometry of the helix as a key factor in the determination of the mechanical properties.

The simulations also allow to extract the propulsive force which the flagellum exerts on the cells. Up to the formation of the screw, the propulsive force $F$ is proportional to the motor torque $M$, i.e., $F = \beta\text{eff} \cdot M$. A larger value of the slope $\beta\text{eff}$ indicates a more efficient transfer of torque into driving force. The data in Fig. 4c show the linear relation between $F$ and $M$ for the cases of a pure FlaB flagellum and admixtures of 20 and 80% FlaA. The inset shows that the slope $\beta\text{eff}$ increases strongly with the FlaA content until about 20% and varies little thereafter. The data indicate a variation of the onset of screw formation depending on the FlaA fraction and show that in a range between 16 and 20% the cells achieve efficient forward propulsion and maintain the ability to form screws at high torque. This is well within the range of proximal segments we determined for wild-type cells (~8% to ~28%; Fig. 4b). With respect to this pronounced variability in length of the proximal FlaA segment within the flagellar filament, the experimental data in concert with the simulations strongly suggest that *S. putrefaciens* forms a highly heterogenous cell population with respect to propulsion and the ability of screw formation. This may not be optimal at the single cell level but distributes the trade-off between torque generation and propulsive efficiency among the population, which allows efficient spreading of at least a fraction of the population under a wide range of conditions.

## Discussion
About half of all flagellated bacteria possess more than a single flagellin-encoding gene, and it was demonstrated for several species that their flagellar filament is, in fact, assembled from more than one building block. However, it is still unclear why so many bacteria have maintained this arrangement. In this study, we showed that the *S. putrefaciens* flagellum is assembled from two different flagellins, FlaA and FlaB, in a spatially organized fashion and explored how this affects flagellar geometry and function. We provided evidence that the particular spatial assembly by two different flagellins benefits cellular motility when considering a wide range of environmental conditions, free swimming under normal and elevated viscosity and spreading through structured environments. We show that a proximal FlaA segment stabilizes the flagellar filaments to provide a compromise between propulsion and the ability to wrap the flagellum around the cell, which benefits motility in structured environments. In *S. putrefaciens*, the length of this proximal stabilizing segment shows a high variability, leading to a heterogenous population well equipped for different environmental conditions. Taken together, our results provide quantitative support for the hypothesis that structural aspects of filament assembly drive the maintenance of multiple flagellin-encoding genes[8].

How do the bacteria form a spatially organized filament? Our study strongly suggests that the arrangement is achieved by sequential production and export of different amounts of FlaA and FlaB, based on individually regulated promoters rather than differences in assembly efficiency as previously proposed for *S. oneidensis*[31]. A very similar spatial pattern of FlaA/FlaB occurs in the flagellum of the latter species (Supplementary Fig. 10), strongly indicating a similar timing of flagellin production and export, which therefore appears to be conserved among *Shewanella* sp.[24,28,31]. A similar sequential flagellin production may underlie the spatial flagellin distributions in other flagellar filaments composed of two flagellins, e.g. in *Campylobacter coli* or *Helicobacter pylori*, while the more complex flagellar filaments of *Caulobacter crescentus* or *Bdellovibrio bacteriovorus*, both produced from six individual flagellin subunits, likely require a more complex regulation[8,12,14,16,32]. However, several important aspects of regulation remain to be elucidated in *S. putrefaciens*: Flagellin expression and production from the *flaA* promoter is not constitutive, and flagellar filaments produced from this promoter never reach normal length (Supplementary Fig. 5m). Furthermore, the length of the proximal flagellar segment also depends on whether *flaA* or *flaB* is expressed from the *flaA* promoter (Fig. 1; Supplementary Fig. 5 and 8). We therefore hypothesize that, in the wild type, at the onset of FlaB production *flaA* expression is restricted or even shut down by a yet unknown mechanism at the transcriptional and posttranscriptional level. Flagellin expression from differentially controlled promoters suggest that *Shewanella* sp. are capable of regulating the composition and properties of the flagellar filament by tuning the amount of incorporated FlaA in response to yet unknown environmental signals. In *S. oneidensis*, production of FlaA has been shown to be under control of the flagellar regulators FlrB and FlrC[28], which may therefore be involved in this regulation. It is also yet unclear which differences between the flagellins determine the different filament morphologies, as FlaA and FlaB are highly conserved. For *S. oneidensis* it has been proposed that the functional difference between FlaA and FlaB is due to four amino-acid residues within the flagellins[31]. Three of these residues are conserved in *S. putrefaciens* FlaA and FlaB, but did not affect swimming upon substitution (Supplementary Fig. 11). Thus, other residues or factors, such as the differences in flagellin glycosylation, account for the difference in filament morphology and function.

Our study shows that an assembly of the flagellar filament by one or two different flagellins affects its mechanistic properties, which has significant consequences for several aspects of flagellar function in different environments. Directional changes of many monopolarly flagellated bacteria are mediated by combined bending of flagellar filament and hook, the structure which joins filament and motor, during the typical run-reverse-flick movement[25,33–35]. Thus, the proximal FlaA segment in *S. putrefaciens* likely has a positive effect on robust directional changes at high and fast swimming at normal viscosity. In contrast, a proximal FlaB segment as in the FlaBA mutant filament rather suppresses directional switches at high angles. The trajectories of this mutant with an enrichment in small turning angles also point towards rather short motor breaks or very short intervals of rotational switching. This may occur more frequently at elevated viscosity, as indicated by the shorter run periods under these conditions, which may be explained by differences in load associated with the composition of the flagellar filament[36,37]. We have previously demonstrated that *S. putrefaciens* may wrap its polar flagellar filament around the cell body in a spiral-like fashion upon CW rotation, and this process starts with an instability of the proximal filament segment at high load[23]. Our experimental and simulation data strongly suggest that the spatial localization of FlaA at the filament's base stabilizes the flagellum, which may be already conferred by the smaller diameter and pitch of the flagellar helix in this region. Spatial distributions with one flagellin primarily localized at the base of the flagellar filament were reported previously for *C. crescentus*[14] and *H. pylori*[16] and suggest that these segments fulfill a similar role in optimizing flagella-mediated swimming in these species.

Beyond monopolarly flagellated *S. putrefaciens*, the screwing mode of movement with the flagellar filament(s) wrapped around the cell body was shown for the lophotrichously flagellated species *Burkholderia sp.* RPE64, *Aliivibrio fischerii*, and *Pseudomonas putida*[38,39] and bipolarly flagellated *Helicobacter suis*[40]. In all cases, full flagellar screw formation significantly slowed down the speed of free-swimming cells. More efficient propulsion by screwing motility was observed when the cells had surface contact[23,39]. In many of their natural environments flagellated bacteria have to move through structured environments, such as sediments, mucus layers or biofilm matrices, where the bacteria constantly are at risk of getting stuck in passages too narrow for cells to pass. In these environments, polarly flagellated bacteria may employ the flagella-mediated screwing motility to escape from traps or to enable the passage through mucus-filled ducts of higher organisms[23,39], however, clear evidence for such a role was missing so far. In this study, we provided a mutant (FlaA-only), which is completely incapable of screwing motility. This strain retains vigorous free swimming, but spreading through soft agar is drastically diminished, demonstrating that this mode of motility very likely gives a significant advantage for moving through structured environments. We expect screwing motility to be similarly important for numerous other polarly flagellated bacterial species in structured environments.

## Methods

**Bacterial strains, growth conditions, and media**. *Shewanella* and *Escherichia coli* strains that were used in this study are listed in Supplementary Table 1. *Shewanella* strains were grown in LB medium at room temperature or 30 °C, *E. coli* in LB medium at 37 °C. Selective media were supplemented with 50 mg ml$^{-1}$ kanamycin and/or 10% (wt vol$^{-1}$) sucrose when appropriate. For the 2,6-diaminoheptanedioic acid (DAP)-auxotroph *E. coli* WM 3064 media were supplemented with DAP at a final concentration of 300 $\mu$M. To solidify media, LB agar was prepared using 1.5% (wt vol$^{-1}$) agar for regular plates and 0.15–0.25% (wt vol$^{-1}$) select agar (Invitrogen) for soft-agar plates.

**Strains and vector constructions**. Gene knock-out and substitution strains of *Shewanella* were constructed by sequential double homologous recombination. Briefly, in-frame deletions were generated by combining approximately 500-bp fragments of the up- and downstream regions of the designated gene. Only a few codons (typically around six) were left in the genome to prevent disruption of regulatory elements for other genes. In-frame insertions were constructed and re-integrated into those deletion strains in basically the same fashion. Substitution of single or multiple nucleotides was done by inserting a modified fragment into the corresponding deletion strain. Regular genetic manipulations of *S. putrefaciens* CN-32 and *S. oneidensis* MR-1 were introduced into the chromosome by double homologous recombination via the suicide plasmid pNPTS-R6K[41]. Plasmids were constructed using standard Gibson assembly protocols[42] and introduced into *Shewanella* cells by conjugative mating with *E. coli* WM3064 as donor. Plasmids and corresponding oligonucleotides are listed in Supplementary Tables 2 and 3, respectively.

The *S. putrefaciens* strains featuring two flagellin genes under the control of the *flaA* promoter were constructed as described above using synthetic flagellin genes (Genescript) that were codon optimized for *E. coli* K-12 to prevent gene loss by homologous recombination with the native flagellin gene (sequences see Supplementary Table 5). The synthetic gene was introduced downstream of the native gene with a copy of the native Shine-Dalgarno sequence of *flaA* and a spacer of 12 random base pairs.

The FlaA-overproduction plasmid was constructed using standard Gibson assembly protocols[42] and introduced into *Shewanella* cells by conjugative mating with *E. coli* WM3064 as donor. Both *flaA* and *fliS* were introduced into pBTOK downstream of the anhydrotetracycline-inducible promoter each with their own AGGAGG Shine-Dalgarno sequence and with a spacer of 12 random base pairs. Gene expression was induced after 1 h of growth in LB with 10 ng ml$^{-1}$ anhydrotetracycline.

**Flagellar filament staining and microscopy**. Flagellar filaments were visualized by exchanging surface-exposed threonine residues of the flagellin monomers to cysteine residues and selectively coupling maleimide-ligated fluorescent dyes to the SH-group of cysteine[23]. To rule out that cysteine substitutions within the flagellin negatively affect the swimming behavior we performed Western Blot analysis, soft agar assays and single cell speed analysis.

For microscopy, cells of an exponentially growing LB culture (OD600 = 0.6) were harvested by centrifugation (1200 × *g*, 5 min, room temperature) and resuspended in 50 $\mu$l PBS. To prevent flagella being sheared off, the pipette tip was

always cut off when pipetting cells. 0.5–1 $\mu$l Alexa Fluor 488 C5 maleimide (Thermo Fisher Scientific) or CF™ 488 maleimide (Sigma-Aldrich) was added and the cell suspension was incubated in the dark for 15 min. Cells were sedimented again and carefully washed with 1 ml of PBS to remove residual unbound dye. After final resuspension in LM100 medium (10 mM HEPES, pH 7.3; 100 mM NaCl; 100 mM KCl; 0.02% (wt vol$^{-1}$) yeast extract; 0.01% (wt vol$^{-1}$) peptone; 15 mM lactate) the cells were kept shaking in the dark until microscopy, but never longer than 30 min. Image sequences of typically 150–300 frames and 20–50 ms exposure time were taken at room temperature using a custom microscope setup (Visitron Systems) based on a Leica DMI 6000 B inverse microscope (Leica) equipped with a pco.edge sCMOS camera (PCO), a SPECTRA light engine (lumencor), and an HCPL APO ×63/1.4–0.6 objective (Leica) using a custom filter set (T495lpxr, ET525/50m; Chroma Technology).

**Determining flagellar screw formation**. To determine the screw formation frequency of the four types of flagellar filament mutants, an aliquot of stained cells was loaded on a swim slide and monitored about 100 $\mu$m away from the glass surfaces. For each filament type movies of three biological replicates were recorded on subsequent days in regular LM medium and LM supplemented with 15 % (wt vol$^{-1}$) Ficoll® 400 (diluted from a 50 % (wt vol$^{-1}$) stock solution). This small, highly branched polymer has been observed to act as a purely viscous contribution to the solvent rheology in experiments with *E. coli*[43]. Several image sequences of 200 frames at 30 ms exposure time were taken for each condition. Screw formation was determined manually for backward swimming cells or cells switching to backward swimming. For each condition about 300 backward swimming events were counted.

**Determining flagellar helix parameters**. To measure the pitch, diameter and axis length of the flagellum the cells were prepared as described above and supplemented with 50 $\mu$M phenamil prior to loading on the swimslide. Phenamil stopped or at least slowed down the rotation of the Na$^+$-driven motor so that images of the helical waveform of the flagella could be obtained. The proportion of the proximal flagellin fragment could only be measured for FlaA, as this flagellin shows brighter fluorescence and could be clearly distinguished from the cell body and the remaining part of the flagellum. Parameters were measured using the ImageJ distribution Fiji[44].

To determine the handedness of the different types of flagellar filaments z-stack image sequences of fluorescently labeled flagella were recorded as described above. Shifting the focal plane through the cell body shows characteristic patterns for left-handed and right-handed helices. The flagellar helices of all flagella types were found to be left-handed.

**Protein separation and western immunoblotting**. For western immunoblotting, cell lysates of exponentially growing LB cultures were obtained by centrifuging cells corresponding to an OD600 = 10 and resuspending the cell pellet in Laemmli buffer[45]. Prior to protein separation by SDS/PAGE using 12% (wt · vol$^{-1}$) polyacrylamide gels the samples were heated to 95 °C for 5 min Subsequently, the proteins were transferred to a nitrocellulose Roti-PVDF membrane (Roth) by semidry transfer. The polar flagellins FlaA and FlaB (of *S. putrefaciens* and *S. oneidensis* alike) were detected with polyclonal antibodies which were raised against the N-terminal conserved region of *S.* MR-1 FlaB (Eurogentec Deutschland) in the dilution of 1:500. As secondary antibody anti-rabbit IgG-horseradish peroxidase (Thermo Fisher Scientific, prod. # 31460) was used at a dilution of 1:20,000. FLAG-tagged FliS was detected with a monoclonal, horseradish-peroxidase-conjugated antibody raised against the FLAG-tag (Sigma Aldrich, prod. # A8592) in the dilution of 1:1000. The horseradish peroxidase signal was detected with the CCD System LAS 4000 (Fujifilm) after incubating the membranes with SuperSignalH West Pico Chemiluminescent Substrate (Thermo Scientific) for one minute. If portions of gels or blots are shown the full gels or blots can be found in Supplementary Figure 12.

**Soft-agar spreading assay**. Spreading ability in complex environments was carried out in soft-agar medium. LB medium supplemented with 0.15–0.25% (wt vol$^{-1}$) select agar (Invitrogen) was heated carefully and cooled to 30–40 °C before pouring the plates. After solidification 2 $\mu$l of exponentially grown cultures were spotted on the plates and incubated in a moist environment over night at room temperature. Plates were scanned before the swim colonies merged using an Epson V700 Photo Scanner. Cells to be directly compared were always incubated on the same plate.

**Statistics of filament parameters and screw formation**. Significance of the relative transcription levels (qRT-PCR) and the filament parameter measurements were tested with a two-sided two-sample *t*-test or Welch's *t*-test ($P < 0.05$, Bonferroni corrected) in R version 3.3.2. Mean, Median and SD were calculated in Microsoft Excel 2013. For each flagellar filament type pitch, diameter, axis length and if possible the proportion of the proximal filament fragment of the flagellum of 50 cells were measured. The actual length of the filament (arc length) was calculated in Excel 2013. Significance of screw formation (only if screws were observed at all) was tested pairwise with two-sided Fisher's exact test ($P < 0.05$, Bonferroni

corrected) in R version 3.3.2. 95% confidence intervals were calculated with the exact binomial test. For each flagellar filament type and condition about 100 backward swimming events were counted on 3 subsequent days, resulting in a total of about 300 counts for each filament type and condition. All measurements and the corresponding statistics are summarized in Supplementary Table 4.

**Isolation of total RNA and qRT-PCR.** Total RNA of exponentially growing *S. putrefaciens* CN-32 cells (OD600 = 0.5, three biological replicates) was extracted using the hot-phenol method[46]. Briefly, the cells were incubated in SDS-acetate buffer (20 mM sodium acetate, 1 mM EDTA, 1% SDS, pH 5.5), with one and a half volumes of hot phenol (pH 4.3) at 65 °C for 10 min, inverting the tube every 2 min. After centrifugation, the aqueous phase was extracted twice using phenol-chloroform-isoamyl alcohol (25:24:1) and RNA was precipitated with 1/8 volume of 2 M sodium acetate solution (pH 5.2) and 2.5 volumes of ice cold ethanol (>96% v v$^{-1}$). After incubation at −20 °C overnight, the pellet was washed with ice cold ethanol (70% v v$^{-1}$), dried and resuspended in RNase-free water. The quality of the RNA was determined by agarose gel electrophoresis and concentration was measured at 260 nm. Residual DNA was removed with the Turbo DNA-free kit (Applied Biosystems) according to the manufacturer's instructions. The RNA extract was then applied for random-primed first-strand cDNA synthesis using BioScript reverse transcriptase (Bioline) according to the manufacturer's instructions. RNA and cDNA samples were stored at −80 °C. The cDNA samples were used as a template for qRT-PCR (C1000 Thermal Cycler with the CFX96 Real-Time System, Bio-Rad) using the SYBR green detection system, MicroAmp Optical 96-well reaction plates and Optica adhesive covers (Applied Biosystems). As *flaA* and *flaB* gene sequences are very similar the specificity of the used oligonucleotides (Supplementary Table 3) was verified by PCR using total DNA extracts (E.Z.N.A. ® Bacterial DNA Kit, Omega) of Δ*flaA* and Δ*flaB* mutants. RNA samples treated without reverse transcriptase were used to test for DNA contaminations in the extracted RNA. PCR amplified DNA fragments of the flagellin locus (using oligonucleotides B 49 and B 50, Supplementary Table 3) were used as positive control and water as negative control. The cycle threshold (Ct) was determined automatically after 40 cycles (Real-Time CFX Manager 2.1, Bio-Rad). Ct values of the flagellin genes were normalized to the Ct values obtained for *gyrA* (Sputcn32_2070). Primer efficiencies and relative transcript levels were determined according to Pfaffl[47] and used to estimate the differences in transcript amounts of the two flagellins.

**Holographic cell tracking.** Cells were grown according to the microscopy protocol above, but without the flagellar staining step. The cells were loaded into glass sample chambers measuring approximately 5 mm × 20 mm × 0.5 mm constructed from glass slides and UV-curing glue. The sample was illuminated using a fiber-coupled laser diode (wavelength $\lambda = 642$ nm) mounted in place of the condenser lens assembly on a Nikon Ti U inverted microscope. The sample was imaged using a standard bright field objective lens (×10, NA 0.3) onto a CMOS camera (Mikrotron MC-1362). Movies were acquired at 50 Hz for around 60 s, at a resolution of 1024 × 1024 pixels. This arrangement gave a sensitive volume of around 1.4 mm × 1.4 mm × 0.5 mm in which to track cells. This resulted in 4887 tracks for the wild type, 4611 tracks for FlaB-only, 1883 tracks for FlaA-only and 4850 tracks for FlaBA in regular medium and 1340 tracks for the wild type, 2558 tracks for FlaB-only, 3019 tracks for FlaA-only and 1623 tracks for FlaBA in high viscosity medium. Note that even the smallest data set adds up to over 5.5 h of recorded and analyzed swim time. The movies were analyzed offline to extract cell coordinates in each frame, and to assemble these coordinates into cell tracks using custom-written software routines[48–51]. Cell tracks were smoothed with piecewise cubic splines to remove noise[52] and analyzed to extract the instantaneous velocity, tumble angle and run time distribution.

The raw data was processed stepwise to obtain cell coordinates. We obtained a background image from each movie sequence by finding the median pixel value at each position in the image. We divided each image in a sequence by this static background, and applied the Rayleigh-Sommerfeld back-propagation approach[48], to produce a stack of numerically refocused images at various distances from the original optical plane. A spatial bandpass filter was applied during the reconstruction to reduce pixel noise; the filter preserved objects between 2 and 30 pixels in size. At our relatively low magnification (×10 total magnification), cells appear point-like in the resulting images. We applied a Sobel-type filter[49,52] to obtain the position of each cell in three dimensions.

To reconstruct cell tracks an extract instantaneous velocities a custom software routine was written to connect the positions of cells between frames in order to assemble three-dimensional tracks. The position resolution carried an uncertainty of approximately ±0.5 μm in the directions perpendicular to the optical axis, and ±1 μm parallel to the optical axis in each frame, at this magnification. All cells were also subject to translational Brownian motion as they swam; the combination of high-frequency pixel noise and Brownian motion complicated efforts to calculate the time-derivatives of position, and therefore the swimming speed. To smooth short-time fluctuations associated with these noise sources, the cell tracks were fitted using piecewise cubic splines. This allowed us to reliably extract instantaneous cell velocities. This technique also allows us to reject the cells that are moving by Brownian alone. The spline-smoothed trajectory of a non-motile particle has a mean-squared displacement that scales with time, whereas a

swimming cell's mean squared displacement scales with time squared. By setting a threshold for the slope of the mean-squared displacement at short times, and a threshold for the total mean squared displacement after 2 s, we can effectively eliminate non-motile particles.

Extraction of tumble angle and run duration distribution was done by calculating the cell's instantaneous direction at each time point, along with the change in direction, $\theta(t)$, between subsequent time points. These data were scanned using a standard peak search algorithm to detect the position of maxima in d$\theta$/dt above a threshold of 5°s$^{-1}$. These maxima were associated with reorientation events; residual effects of Brownian motion after the spline smoothing are sufficient to allow pauses followed by a resumption of swimming (equivalent to a reorientation angle near zero degrees) to be detected as reorientation events. Example data are shown in Supplementary Figure 13; 13a shows a single cell track (wild type, the same track shown in Fig. 3), and 13b displays a graph of d$\theta$/dt against time in which the events noted as tumbles are highlighted with blue circles. The total change in swimming direction at each re-orientation event was calculated by comparing a cell's swimming direction 0.25 s before a maximum in $\theta(t)$ with the swimming direction 0.25 s after the maximum. The time between subsequent re-orientation events was used to build a distribution of run durations. Only tracks with two or more re-orientation events contributed to the run duration distributions.

**Numerical model for the flagellar filament.** The numerical model for the flagellar filament is based on on that proposed by Vogel and Stark[53], and the same as the one used in Kühn et al.[23]. It is based on a representation of the flagellum as a set of beads with mutual interactions such that they form a helix with prescribed radius $R$ and pitch $P$ at rest. The differences between FlaA and FlaB are reflected in the local radius $R$ and pitch $P$ of the helix: FlaA forms narrow helices with $R = 0.175$ μm and $P = 1.18$ μm, whereas FlaB forms wider helices with $R = 0.315$ μm and $P = 1.91$ μm. The torque-force relation is obtained by direct numerical integration of the equations of motion at fixed torque, and then by computing the mean force over 10 rotation periods to average out the oscillations that are connected with the positioning of the motor relative to the axis of the filament. As in the previous simulations[23], we observe that polymorphic transitions within the filament reduce the required torque, and thus have a quantitative but not a qualitative effect on the changes in mechanical properties[53,54].

With s the arclength along the rod, the torsional state of the rod is described by the rotational strain vector $\mathbf{\Omega}(s)$. Deviations from the rest state give quadratic contributions to a free energy

$$\mathcal{F}_K = \int_0^L \left(\frac{A}{2}\right)\left[\left(\Omega_1 - \Omega_{0,1}\right)^2 + \left(\Omega_2 - \Omega_{0,2}\right)^2\right] + \left(\frac{C}{2}\right)\left(\Omega_3 - \Omega_{0,3}\right)^2 ds, \quad (1)$$

where $A$ is the bending and $C$ the torsional rigidity.

Polymorphism is introduced by calculating the free energy relative to the different equilibrium states and taking the minimum: if the flagellum is stretched too far from one local minimum it can fall into the basin of attraction of another one and relax to a different configuration. For FlaB we find two relevant equilibrium configurations, but for FlaA there is no evidence for a second one. We take the same elastic coefficients $A$ and $C$ for all minima, since we do not have independent numbers for the different equilibrium states (neither for the polymorphic states nor for FlaA vs FlaB).

In addition to the torsional contribution the free energy also contains a global harmonic spring potential

$$\mathcal{F}_S = (K/2)\int_0^L (\partial \mathbf{r}/\partial s)^2 ds \quad (2)$$

with an elastic constant $K$ that keeps the variations in length within 0.1%.

The interaction between flagellum and fluid is calculated within resistive force theory[55]. There are three local friction coefficients that depend on the geometry of the flagellum:

$$\gamma_\perp = \frac{4\pi\eta}{\ln(0.09l/r_f) + 1/2} \quad (3)$$

is the friction coefficient for motion perpendicular to the centerline, with $\eta$ the viscosity, $r_f$ the radius of the filament and $l = \sqrt{4\pi^2 R^2 + P^2}$ the contour length of one helical turn. The tangential friction coefficient is

$$\gamma_\parallel = \frac{2\pi\eta}{\ln(0.09l/r_f)} \quad (4)$$

and the rotational coefficient is

$$\gamma_r = 4\pi\eta r_f^2 \quad (5)$$

As in Vogel and Stark and Kühn et al.[23,53], the equations of motion are integrated with an embedded Cash–Karp method[56], and determine the positions and velocities of the attached tripoid following the procedure proposed by Chirico

and Langowski[57]. The runs are started with a left-handed helix with $P^S = 1.91\,\mu m$ and $R^S = 0.315\,\mu m$ for segments containing FlaB, and with $P^S = 1.18\,\mu m$ and $R^S = 0.175\,\mu m$ for segments containing FlaA. The contour length of the flagellum was set to $Lc = 6\,\mu m$, corresponding to 2 to 4 helical turns, depending on the FlaA to FlaB ratio. The various parameters in the model for the flagellum and their values are summarized in Supplementary Table 6.

**Code availability**. The full code for the three-dimensional tracking has been published in the corresponding references[49,52]. The full code for simulations of flagellar screw formation is available upon request from the corresponding authors.

## Data availability

The tracking data are available at the York Research Database (https://doi.org/10.15124/675b6083-2a6e-43d1-8511-b7c18bb1be9a). Other data supporting the findings of the study are available in this article and its Supplementary Information files, or from the corresponding authors upon request. A reporting summary for this Article is available as a Supplementary Information file.

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

## Acknowledgements

M.J.K. and F.K.S. were supported by grants TH831/6-1 and BE102/14-1, respectively, from the Deutsche Forschungsgemeinschaft within the framework of the priority program 1726. N.E.F. and L.G.W. were supported by EPSRC grant no. EP/N014731/1. We are grateful to Ulrike Ruppert for technical support.

## Author contributions

M.J.K., F.K.S., L.G.W., B.E., and K.M.T. conceived the project and designed experiments; M.J.K., F.M.R., and B.H. constructed strains, M.J.K. performed single-cell microscopic analysis and regulation studies, N.E.F. and L.G.W. performed holographic cell tracking, F.K.S. and B.E. provided numeric simulations. M.J.K., L.G.W., F.K.S., B.E., and K.M.T. wrote the paper. All authors discussed the results and commented on the manuscript.

## Additional information

**Competing interests:** The authors declare no competing interests.

