## [Peer Review File · Nature Communications]

Reviewers' comments:

Reviewer #1 (Remarks to the Author):

In this article the authors explore the role of the two different flagellins FlaA and FlaB in *Shewanella putrefaciens*. They fluorescently label the FlaA and FlaB separately and showed that in the wild type bacteria, FlaA is present in the first 20% of the flagellum, and FlaB predominantly composes the remainder of the flagellum. Furthermore mutants without FlaB only produced short stubs of flagella, while mutants without FlaA produced flagella with normal morphology. Investigating the regulation of flagellin production, it was shown that production of both flagellins requires RpoN and flrA regulators, but fliA is required only for production of FlaB. Since the flaB gene is located after the flaA gene, they hypothesized that FlaB is only produced after the hook is finished, while FlaA is always present and forms a pool which initially builds the basal section of the flagellum. Switching the positions of flaA and flaB so that flaA is controlled by the flaB promoter switched the led to a full FlaA-only flagellum for mutants without flaB, and to a flagellar stump of FlaB for mutants lacking flaA. Image analysis revealed that FlaA flagella have a smaller helical radius than FlaB or wild-type flagella.

Next the authors investigated the swimming characteristics of FlaA-only, FlaB-only, and wild type flagella. In an agar spreading assay, cells with FlaB-only and wild-type flagella spread similarly, while those with FlaA-only flagella only spread about half as much. By analyzing swimming trajectories imaged by 3D holographic tracking, it was determined that all three strains had similar speeds. Each strain split into two subpopulations, one with slow and one with fast swimming. In the fast subpopulation the wild-type had the highest speed and with the FlaA- and FlaB-only slightly slower. The size of the slow subpopulation increased dramatically as viscosity was increased in wildtype and FlaB-only strains, while it remained the same size for FlaA-only strains. Using fluorescence imaging the slow subpopulation was associated with "screwlike" flagellar configuration wrapping around the body that is advantageous for transport and spreading in structured environments; hence screw formation was stopped in FlaA-only strains. Finally, FlaA- and FlaB-only strains showed reduced ability to "flick" effectively, especially at high viscosities, while wild-type cells were able to "flick" in low- and high-viscosity conditions.

The effect of the flagellins on the ability to form the screw configuration was tested in numerical simulations that treated flagella as beads connected to each other by elastic springs with preferred bonding angles that mimic the different helical geometries produced by the flagellins. Hydrodynamics are treated by resistive force theory. It was found that the critical motor torque needed to form a screw occurred at about 5 pN um for FlaB flagella, and that the critical motor torque increased to 8.5 pN um as the first 20% of the flagellum was replaced by FlaA, after which it decreases until 40% of the flagellum is FlaA, and then increases again. Furthermore, the thrust force per unit torque produced by the flagellum increases rapidly as the first 20% of the flagellum is replaced by FlaA, and then increases very slowly for more FlaA.

The authors conclude that the observed mixture of 20% FlaA flagellum stabilizes the flagellum against screw formation but still allows screw formation in structured environments when it is beneficial, while also maximizing the propulsive force of the flagellum.

In my opinion, the overall idea of the paper is quite interesting, that the production and location of the two flagellins in flagella is regulated to control the behavior of the flagellum in its various environments. However, there are enough questions and concerns I have about the specific conclusions made in this case (and detailed below) that I can't recommend publication in the current form.

Major concerns

First I am not sure that the experiments have thoroughly checked all the hypotheses proposed in the manuscript:

1) Regarding the spatial regulation of FlaA and FlaB into the flagellum, I am uncomfortable with some of the language stating that they showed (line 315) that the arrangement is achieved by sequential production of the two flagellins. Earlier they are more circumspect and say a potential mechanism is "suggested". To say the scheme in Fig S4 is "showed" I would expect some more ironclad experimental tests of the mechanism proposed. For example, an obvious question that comes to mind is what happens in the case when the locations of flaA and flaB are switched, but both are present? Is the basal 20% of the flagellum FlaB, and the remainder FlaA, as their mechanism suggests it should be? I'm not sure why this wasn't tested or reported. In addition, the work described still doesn't really explain a) what regulates only the first 20% of the flagellum to be FlaA, why not some other amount? b) Why with FlaA-only, there is a stump -- shouldn't FlaA continue to be produced, and eventually grow to a longer flagellum?

2) Regarding the morphology of FlaA flagellum and how it inputs into the model as affecting the geometry of the flagellum. Based on the statements in the manuscript, I would expect that the wild-type flagellum should display the FlaA morphology (smaller radius) for the first 20% and then switch to the larger radius in the rest of the flagellum. Is that observed in the fluorescence imaging of the flagella? This would help to justify the idea that only the geometry of the flagellum is driving the different behaviors, rather than things like the

Major concerns, continued

Second I find myself quite unsatisfied by some of the implied evolutionary arguments explaining the function of the two flagellins and their arrangements:

3) First, because they are implied rather than explicit I am not sure what they are. Do the authors believe that motility in structured environments is driving the regulation of these flagellins? Or is free swimming? Or perhaps both, and the ability to do both.

4) There seems to be a circular aspect to their conclusions in the sense that it is a just-so story, where what is observed behaviorally is attributed to being what drives the regulation of the flagellins. The most succinct argument for the observed arrangement is that it allows the flagellum to achieve screw formation in the "best" critical torque range while making efficient torque generation. Why is that the "best" range? Could there be some way to really test this just-so story instead of just postulating it? Even within the story, there seems to me to be the following unexplained fact -- based on the results in Fib 3b and 3c, at about 50% FlaA, there is the same critical torque, and beta_efficiency is in fact better than 20% FlaB. Why is this not the observed flagellin arrangement instead?

Minor comments:

5) I was confused by lines 229-231. the statements seemed contradictory as to whether the strains performed flicks or not, as well as which strains (FlaA and wild type, or FlaA- and FlaB-only) tended to turn at high angles.

6) It is mentioned that polymorphism is important in the screw formation, and that FlaB is assigned two stable polymorphic forms in the model. However, no explanation of the experiments done to lead to those conclusions is included -- which should be.

Reviewer #2 (Remarks to the Author):

I want to support this manuscript as there are a number of things to like about it. First the authors provide a potential reason for cells to encode multiple flagellin monomers simultaneously and a phenotypic consequence if they can't. Specifically, they show that cells that make a filament out of FlaA filament protein alone are unable to have the flagellum wrap around the cell body and rotate to propel the cell like a screw. I don't think all bacteria with a single polar flagellum can use the screw-like movement and this work may also explain why that is, their flagellin won't permit it. I feel that the most important and most clearly presented observation has to do with how secretion is regulated somehow at the transcriptional level. Moreover, the transcription based secretion control, while mysterious and intriguingly counter-intuitive, also helps explain the localization observation that the flagellin expressed prior to the substrate specificity switch comes to predominantly occupy the base of the filament. They further show that filaments made from one protein or the other have different behavioral parameters. How and why flagellar filaments are assembled from multiple different monomers is important, and the screw like behavior observed in *Shewanella* feels like it is important for the mechanism of spirochaete force generation.

The assembly and localization work is very nice and clear. Once the paper switched to the behavior analysis, I became completely lost. It was hard to understand what the phenotype of the different flagellar filaments were and which differences were relevant. It was further difficult to understand whether the differences were consistent with the mechanistic model or whether defects in one assay could explain defects in another. A computational model is provided but it was poorly described and used as proof of the mechanism. I will never accept computational simulations as proof of anything biological. Instead, the authors could consider moving the simulations forward to predict possible behavior outcomes that are then experimentally tested (i.e. move fig 3b,c before fig 2, I think all Fig 3b,c needs is the pitch measurements of the uniform filaments in Fig 1). As written, it isn't even clear which aspects of the data the simulations support and which, if any, they do not support.

Finally, the authors want to conclude that spatial localization of the different flagellins is important for behavior but they don't actually test the connection between localization and behavior. Later, they want to conclude that the different ratios of one flagellin to another is important for behavior but they don't test that either and it isn't clear what the ratios actually are. When the filament is made entirely on one kind of flagellin, behavior is altered but again, I can't clearly articulate with certainty what the defect is or how it is related to the flagellins.

Specific comments

Line 46. The authors don't actually show that the spatial arrangement of the flagellins is responsible for screw-like behavior. Instead, they show that the presence of FlaB is necessary for the screw like behavior. To test spatial importance I believe they would have had to generate a strain in which a strain expressed both flagellins, each under control of the opposite promoter, that FlaB now occupies the proximal zone, and that the remainder of the FlaA filament isn't sufficient to promote screw-like movement. The combination of flagellin expression I propose is the only one missing from Figure 1.

Lined 55. Slot "favorite" for "favorable". Bacteria don't have favorites.

The section starting line 167 is very interesting. I agree with the authors interpretation that the first flagellin in the tandem array is expressed before the second, FliA-dependent flagellin and the substrate specificity switch and this preloads the cell with the proximal subunit. But I think what the data also says is that the first flagellin secretion stops immediately after the substrate specificity switch has been flipped. Has it ever been demonstrated as clearly as this that once the switch to late class flagellar proteins is made, that early class flagellar proteins are no longer

secreted? Else, why does elongation stop? I believe current models would say otherwise that the chaperones control preferential late class protein secretion and outcompete, but not abolish, the early class. Very strange. Moreover, these data suggest something fundamental about T3SS specificity. If I'm understanding the constructs correctly, the only thing that differs is the promoter? How would the promoter control secretion specificity? Or does activation of late class gene expression somehow completely inhibit early class? In short, please provide some explanation for why stubs occur when either protein is expressed from the early promoter.

Paragraph starting 211. This paragraph was difficult to read and understand. Perhaps this can be improved by including figure citations at appropriate places between line 211 and 235.

Fig 2a-f. What is P on the X-axis? Percentage?

Fig 2c-d. Why do some cells swim 10-fold faster than others? Are the "slow swimming cells" non-motile and moving by Brownian motion? At first I thought the slow moving cells were the screw like movements but then the FlaA only strain still has a slow moving subpopulation and the authors conclude that it can't move like a screw.

Line 199. Clarify. Is this sentence referring to panel n?

Line 228. Omit "as expected" as it is not clear why this was expected.

Line 233. Clarify "this effect" as I'm not sure what effect is being described. Could this entire paragraph be rewritten for clarity? I've reread it three times and I don't have a clear idea of what the phenotype is. For instance, line 229 seems to indicate that FlaA-only cells are like wild type but line 231 seems to indicate that FlaA-only cells are like FlaB-only cells.

Paragraph 236. I don't understand this paragraph either. It says that single flagellin filaments have significant effects on swimming behavior and then in the next sentence says that FlaA-only mutants have no behavioral phenotype?

Line 245. "The pronounced increase in slow swimming cells for the wild type" seems a strange way to frame the data. Shouldn't this be "the pronounced decrease in slow swimming cells for the FlaA-only strain" as the comparison is how the mutant differs from the wild type? Moreover, I don't understand the conclusions of this section. The biophysical analysis in figure 2d indicates that FlaA-only cells perform better in high viscosity and I think the model is that the screw behavior is advantageous at high viscosity but Fig 3a indicates that FlaA only cells are unable to conduct screw behavior. If the plate assay motility simulates microviscosity environments (dead ends) and the defect of the FlaA-only cells (Fig 1n) is the inability to adopt a screw, why then does FlaA-only overperform in high viscosity (Fig 2d)?

There needs to be an explicit statement as to how motility plates, viscosity, slow and fast swimming subpopulation and torque are related to each other. Otherwise, the paper reads somewhat like a shell game where phenotypes are revealed but you can't tell whether it was in any way related to a previously revealed result.

Line 258. "with wrapped-up flagella" is this jargon that is synonymous with "screw-like motion"?

Line 259. "This swimming phenotype", clarify, there are many parameters being discussed.

Line 258, Line 262. "Not a single" if not a single cell was observed, how is there a standard deviation in the graphs? Moreover, why is the comparison of two "not a single" result being clarified as "n.s." not significantly different in Fig 3a? By definition, identical results are not significantly different.

Paragraph starting line 265. I have no idea what is going on in this paragraph, what figure 3b is, or how it relates to the data. I believe it to be a mathematical model that supports the need for FlaB to cause the flagellum to wrap around the body of the cell but the text supporting it was very hard to understand.

Paragraph starting 280. I'm not really sure what this paragraph is about but I think it is more mathematical modeling?

Line 292. Mathematical simulations don't confirm anything. At best they support the biological observations.

Is the 20% FlaA composition supported by the protein data where FlaA and FlaB can be distinguished by size in an SDS PAGE gel? FigS2a shows different ratios by immunoblotting which could be complicated by glycosylation altering antigen access. I believe there is also a coomassie stain located below each immunoblot (not explained in the figure legend). In the coomassie stain of panel A, there are bands that could be equal intensity?

Reviewer #3 (Remarks to the Author):

This manuscript describes the role of multiple flagellins in the motility of *Shewanella*. Ignoring the presence or absence of a sheath, flagellated bacteria can (roughly) be placed into two categories. There are the enteric bacteria exemplified by *E. coli* & *Salmonella*. They express multiple flagella composed of a single flagellin type. True, most *Salmonella* have antigenically distinct flagellins, but they are functionally identical. Historically, these two bacteria have served as model systems for flagellar function, assembly, coupled gene regulation and chemotaxis. However, there is a whole other world of bacteria that have flagella composed of different flagellin types and the top of the regulatory hierarchy is completely different than the enteric FlhDC-control type. *Shewanella* is an excellent example of non-enteric bacteria that produce multiple flagellin types because it has only two subunit types making it easier to characterize than those with many more.

The authors first demonstrate that the unexpectedly large masses of FlaA and FlaB on gels is due to glycosylation and that glycosylation is lost in deletions of *S. oneidensis* flagellin modification orthologs *pseG* or *maf-1*. The authors need to explain why deletion either gene gives the same result. Are the sequential steps in a known glycosylation pathway?

The authors then demonstrate the spatial arrangement of FlaA and FlaB with FlaA polymerized at the filament base. It is clear that cells transition from FlaA to FlaB since *flaB* deletions produce short FlaA-filaments suggesting a mechanism in place that limits FlaA secretion. The authors then go on to show that FlaA (and FlhS) are produced early in the regulatory cascade before FlhA-dependent FlaB production, which nicely allows coordination of gene expression with FlaA-first assembly. Also, expressing *flaA* from the *flaB* locus and vice versa demonstrated that the mechanism controlling short FlaA filament length was at the level of gene expression. Nice!

The authors go on to get at the reason for different flagellins by measuring numerous parameters - velocity, run durations, turning angles - under different environmental conditions - normal medium versus high viscosity (10% Ficoll). The increase in slow-swimming cells for the wild-type and FlaB-only mutant led the authors to test the effect of high load on filament instability and the screw-like slow mobility. This makes sense for organisms that have exposure to changing environmental conditions - liquid versus solid surface contact, sediments, etc., unlike enterics that live in a more homogeneous viscous environment. Thus, this represents a relatively comprehensive study, which provides a solid evidence for their hypothesis as to the need for multiple flagellin types requires for a lifestyle in the real (non-enteric) world. Great paper!

Minor comments:

line 55: change "favorite" to "optimal" - how do you know what their "favorite niche is? Their true favorite niche might be a bar in Berlin.

line 59: add "subunits" after "flagellin"

line 61: change "therefore" to "their expression is"

line 118: "several"??? three or four??? I need a more accurate number here

Kühn et al., 2018; Answers to the Reviewers

Reviewer #1 (Remarks to the Author):

In this article the authors explore the role of the two different flagellins FlaA and FlaB in *Shewanella putrefaciens*. They fluorescently label the FlaA and FlaB separately and showed that in the wild type bacteria, FlaA is present in the first 20% of the flagellum, and FlaB predominantly composes the remainder of the flagellum. Furthermore mutants without FlaB only produced short stubs of flagella, while mutants without FlaA produced flagella with normal morphology. Investigating the regulation of flagellin production, it was shown that production of both flagellins requires RpoN and FliA regulators, but FliA is required only for production of FlaB. Since the *flaB* gene is located after the *flaA* gene, they hypothesized that FlaB is only produced after the hook is finished, while FlaA is always present and forms a pool which initially builds the basal section of the flagellum. Switching the positions of *flaA* and *flaB* so that *flaA* is controlled by the *flaB* promoter switched the led to a full FlaA-only flagellum for mutants without *flaB*, and to a flagellar stump of FlaB for mutants lacking *flaA*. Image analysis revealed that FlaA flagella have a smaller helical radius than FlaB or wild-type flagella.

Next the authors investigated the swimming characteristics of FlaA-only, FlaB-only, and wild type flagella. In an agar spreading assay, cells with FlaB-only and wild-type flagella spread similarly, while those with FlaA-only flagella only spread about half as much. By analyzing swimming trajectories imaged by 3D holographic tracking, it was determined that all three strains had similar speeds. Each strain split into two subpopulations, one with slow and one with fast swimming. In the fast subpopulation the wild-type had the highest speed and with the FlaA- and FlaB-only slightly slower. The size of the slow subpopulation increased dramatically as viscosity was increased in wildtype and FlaB-only strains, while it remained the same size for FlaA-only strains. Using fluorescence imaging the slow subpopulation was associated with "screwlike" flagellar configuration wrapping around the body that is advantageous for transport and spreading in structured environments; hence screw formation was stopped in FlaA-only strains. Finally, FlaA- and FlaB-only strains showed reduced ability to "flick" effectively, especially at high viscosities, while wild-type cells were able to "flick" in low- and high-viscosity conditions.

The effect of the flagellins on the ability to form the screw configuration was tested in numerical simulations that treated flagella as beads connected to each other by elastic springs with preferred bonding angles that mimic the different helical geometries produced by the flagellins. Hydrodynamics are treated by resistive force theory. It was found that the critical motor torque needed to form a screw occurred at about 5 pN μm for FlaB flagella, and that the critical motor torque increased to 8.5 pN μm as the first 20% of the flagellum was replaced by FlaA, after which it decreases until 40% of the flagellum is FlaA, and then increases again. Furthermore, the thrust force per unit torque produced by the flagellum increases rapidly as the first 20% of the flagellum is replaced by FlaA, and then increases very slowly for more FlaA.

The authors conclude that the observed mixture of 20% FlaA flagellum stabilizes the flagellum against screw formation but still allows screw formation in structured environments when it is beneficial, while also maximizing the propulsive force of the flagellum.

In my opinion, the overall idea of the paper is quite interesting, that the production and location of the two flagellins in flagella is regulated to control the behavior of the flagellum in its various environments. However, there are enough questions and concerns I have about the specific conclusions made in this case (and detailed below) that I can't recommend publication in the current form.

We are grateful for the generally positive support of this Reviewer as well as for pointing out in a highly constructive fashion items that needed further verification. The issues listed will be addressed below.

Major concerns

First I am not sure that the experiments have thoroughly checked all the hypotheses proposed in the manuscript:

1) Regarding the spatial regulation of FlaA and FlaB into the flagellum, I am uncomfortable with some of the language stating that they showed (line 315) that the arrangement is achieved by sequential production of the two flagellins. Earlier they are more circumspect and say a potential mechanism is "suggested". To say the scheme in Fig S4 is "showed" I would expect some more ironclad experimental tests of the mechanism proposed. For example, an obvious question that comes to mind is what happens in the case when the locations of *flaA* and *flaB* are switched, but both are present? Is the basal 20% of the flagellum FlaB, and the remainder FlaA, as their mechanism suggests it should be? I'm not sure why this wasn't tested or reported. In addition, the work described still doesn't really explain a) what regulates only the first 20% of the flagellum to be FlaA, why not some other amount? b) Why with FlaA-only, there is a stump -- shouldn't FlaA continue to be produced, and eventually grow to a longer flagellum?

We are truly grateful for this remark and the suggestions for additional experimental support. Accordingly, we conducted several more experiments, which are now included in the manuscript, and adjusted the statements to reflect their outcome where appropriate.

*First of all, to show that there is a difference between *flaA* and *flaB* expression levels, we performed q-RT-PCR (see Figure S7b,c), and by overexpressing *flaA* (in concert with chaperone-encoding *fliS*) in a Δ *flaAB*₁ Δ *flaAB*₂ strain we demonstrate that the expression level directly affects the length of a FlaA-only flagellar filament. As requested by this reviewer, we constructed a strain with a swap of *flaA* and *flaB* (*FlaBA*), which we then extensively tested with respect to filament morphology and swimming behavior during free swimming or expansion through soft agar. As predicted in the 'sequential-production model', in this strain FlaB forms the base of the filament while the residual segment is consisting of FlaA. However, the amount of FlaB is lower than 20%.*

*This leads to the question of what actually regulates the amount of the flagellin expressed from the *flaA* promoter, and this Reviewer has raised a very important issue here.*

First of all: we have added more experimental data which now showed that the amount of FlaA making up the proximal flagellar segment is not sharp-cut, but scatters at around 17 % of the full filament with quite some variation. This point is rather important as this implies a significant heterogeneity with respect to filament stability and, with that, swimming and screwing behavior. We have now added this more explicitly to the manuscript (line 142 ff; line 412 ff, see also the distribution plots added to Fig. 3b and S10c and d). This fact will also come up in our answers to questions 3 and 4 of this Reviewer.

*The reviewer is also right that, if flagellin production from the *flaA* promoter would be constitutive, one would expect longer filaments, in particular in older cells, as it can be observed that FlaA-only filaments can grow to normal length (when produced from the *flaB* promoter, Fig. 1c,g) or even longer (when artificially overexpressing *flaA* as mentioned above (Fig. S7b,c)). We therefore propose that flagellin production from the *flaA* promoter is restricted, or even shut down, upon activation of the *flaB* promoter by *FliA*. However, it remains unclear how this is achieved and why expression of *flaB* from this promoter leads to even shorter stumps. We have tested various obvious possibilities (e.g., repression of the *flaA* promoter by *FliA* upon release), however, we could so far not elucidate the exact mechanism. We have amended the Discussion section (line 341 ff) accordingly. However, although this aspect of regulation remains obscure so far, all old and new results show that sequential production is by far the most likely process by which the cells achieve the spatial flagellin arrangement within the filament, and the general conclusions of this study remain unaffected.*

2) Regarding the morphology of FlaA flagellum and how it inputs into the model as affecting the geometry of the flagellum. Based on the statements in the manuscript, I would expect that the wild-type flagellum

should display the FlaA morphology (smaller radius) for the first 20% and then switch to the larger radius in the rest of the flagellum. Is that observed in the fluorescence imaging of the flagella? This would help to justify the idea that only the geometry of the flagellum is driving the different behaviors, rather than things like the

In wild type cells, the FlaA segment at the base of the flagellum contributes less than 20% of the flagellum's full length. This also constitutes less than a full period of the helix, so the uncertainties in measurements of helix radius are unacceptably large in wild type cells. However, in addressing point 4) of this Reviewers comments (see below), we constructed a FlaAAB strain in which the flaA gene is introduced into the chromosome as a second copy to increase the amount of FlaA at the filament's base. Using this mutant we could show that, in fact, the proximal segment increases in length and exhibits the smaller diameter characteristic for FlaA-only filaments, while the residual filament assumes the architecture of a FlaB-only filament (see Suppl Figure 9c).

Major concerns, continued

Second I find myself quite unsatisfied by some of the implied evolutionary arguments explaining the function of the two flagellins and their arrangements:

3) First, because they are implied rather than explicit I am not sure what they are. Do the authors believe that motility in structured environments is driving the regulation of these flagellins? Or is free swimming? Or perhaps both, and the ability to do both.

We thank this reviewer for this question which led us to clarify our intentions. We believe that the variation in proximal FlaA-incorporation into the filament results in a population exhibiting a wide heterogeneity with respect to filament stability, giving rise to a variety of swimming properties (see line 412 ff), so that at least a subpopulation of the cells will be well equipped for any given condition, be it in structured environments or during free swimming. The regulation of filament composition by two different promoters would, in theory, allow to adapt the filament properties to environmental conditions, however, it remains to be shown if this is the case.

4) There seems to be a circular aspect to their conclusions in the sense that it is a just-so story, where what is observed behaviorally is attributed to being what drives the regulation of the flagellins. The most succinct argument for the observed arrangement is that it allows the flagellum to achieve screw formation in the "best" critical torque range while making efficient torque generation. Why is that the "best" range? Could there be some way to really test this just-so story instead of just postulating it? Even within the story, there seems to me to be the following unexplained fact -- based on the results in Fig 3b and 3c, at about 50% FlaA, there is the same critical torque, and beta_efficiency is in fact better than 20% FlaB. Why is this not the observed flagellin arrangement instead?

We thank this Reviewer for the comments on this point, and we have revised the wording to better articulate our findings. As the reviewer rightly suggests, we find that the wild-type configuration with its variation in the length of the proximal FlaA segment achieves a trade-off between propulsive efficiency (which we assume is what the reviewer means by 'efficient torque generation', as we have not manipulated the flagellar motor) and the ability to form a screw. This is the main conclusion of our experiments, rather than a postulate.

We demonstrate with a study of the mutants what would happen if things were not 'just-so' in our system. Figs. 1k and 1l shows that the FlaB-only mutant marginally outperforms the wildtype when spreading in a structured environment; Fig. 2c shows that FlaA-only mutants have marginally fewer slow-swimming cells than wildtype in low-viscosity media; but both mutants significantly underperform elsewhere (FlaA-

only in agar and FlaB-only in low-viscosity media). The simulations provide a mechanical explanation for the observed phenotypes, allowing us to explore the full gamut of flagellar composition and motor torque. Unfortunately, the variation in flaA expression did not allow a level of detail to fully verify the outcome of the simulation in in vivo experiments. However, in order to corroborate the results of the simulation, we have now included a mutant (FlaAAB) that produces flagellar filaments in which the length of the FlaA-formed proximal segment is increased to about 22%, which better covers the area of increased stability which was suggested by the simulations. Accordingly, we found that the stability of the corresponding flagellar filaments was significantly increased (Fig. S10a).

Taken together (and as mentioned above), we propose that, instead of settling on fixed levels of FlaA, the rather wide variation in the length of the FlaA segment gives rise to a heterogeneous population with different abilities in swimming and spreading through structured environments, which may not be optimal but distributes the trade-off between torque-generation and propulsive efficiency.

Minor comments:

5) I was confused by lines 229-231. the statements seemed contradictory as to whether the strains performed flicks or not, as well as which strains (FlaA and wild type, or FlaA- and FlaB-only) tended to turn at high angles.

We thank this Reviewer for pointing that out. The whole section has been rephrased - also in response to remarks of Reviewer 2, and to accommodate the FlaBA mutant strain.

6) It is mentioned that polymorphism is important in the screw formation, and that FlaB is assigned two stable polymorphic forms in the model. However, no explanation of the experiments done to lead to those conclusions is included -- which should be.

In order to stay as close to the original model which we used to simulate the screw formation in our previous work (see reference 23), we only adjusted the helical parameters in the FlaA and FlaB regions. The inclusion of a second structure was part of the original model, where it lowered the thresholds for screw formation, and was therefore kept. To avoid confusion, this remark has been moved to the Experiments and Methods section, which now includes this explanation.

Reviewer #2 (Remarks to the Author):

I want to support this manuscript as there are a number of things to like about it. First the authors provide a potential reason for cells to encode multiple flagellin monomers simultaneously and a phenotypic consequence if they can't. Specifically, they show that cells that make a filament out of FlaA filament protein alone are unable to have the flagellum wrap around the cell body and rotate to propel the cell like a screw. I don't think all bacteria with a single polar flagellum can use the screw-like movement and this work may also explain why that is, their flagellin won't permit it. I feel that the most important and most clearly presented observation has to do with how secretion is regulated somehow at the transcriptional level. Moreover, the transcription based secretion control, while mysterious and intriguingly counter-intuitive, also helps explain the localization observation that the flagellin expressed prior to the substrate specificity switch comes to predominantly occupy the base of the filament. They further show that filaments made from one protein or the other have different behavioral parameters. How and why flagellar filaments are assembled from multiple different monomers is important, and the

screw like behavior observed in *Shewanella* feels like it is important for the mechanism of spirochaete force generation.

We thank this reviewer for the supportive remarks and for the insightful comments, which we address in the following.

The assembly and localization work is very nice and clear. Once the paper switched to the behavior analysis, I became completely lost. It was hard to understand what the phenotype of the different flagellar filaments were and which differences were relevant. It was further difficult to understand whether the differences were consistent with the mechanistic model or whether defects in one assay could explain defects in another. A computational model is provided but it was poorly described and used as proof of the mechanism. I will never accept computational simulations as proof of anything biological. Instead, the authors could consider moving the simulations forward to predict possible behavior outcomes that are then experimentally tested (i.e. move fig 3b,c before fig 2, I think all Fig 3b,c needs is the pitch measurements of the uniform filaments in Fig 1). As written, it isn't even clear which aspects of the data the simulations support and which, if any, they do not support.

Finally, the authors want to conclude that spatial localization of the different flagellins is important for behavior but they don't actually test the connection between localization and behavior. Later, they want to conclude that the different ratios of one flagellin to another is important for behavior but they don't test that either and it isn't clear what the ratios actually are. When the filament is made entirely on one kind of flagellin, behavior is altered but again, I can't clearly articulate with certainty what the defect is or how it is related to the flagellins.

We agree with this Reviewer that the description of free-swimming behavior and the differences of wild-type and mutant cells required clarification. We have therefore reworked the whole section and hope that this has become clearer now. The short version is: altering the composition of the flagellar filaments leads to significant differences in all kinds of free-swimming behavior, which, however, do not correspond well to spreading abilities through the structured environment of a soft-agar plate. While we fully agree with this Reviewer that the simulation does not provide a proof for our findings, we are convinced that the simulations of the mechanical properties of segregated filaments provide support for the ideas about why it may be an advantage to arrange the flagellins in the flagellar filament in the spatial distribution we find in the wild type. As the findings fit quite well with our experimental data, we see this as a valuable complement to our story, and we have therefore not rearranged the manuscript. The part has also been rephrased for clarity and to avoid the impression that the simulations are used to 'prove' our results.

Specific comments

Line 46. The authors don't actually show that the spatial arrangement of the flagellins is responsible for screw-like behavior. Instead, they show that the presence of FlaB is necessary for the screw like behavior. To test spatial importance I believe they would have had to generate a strain in which a strain expressed both flagellins, each under control of the opposite promoter, that FlaB now occupies the proximal zone, and that the remainder of the FlaA filament isn't sufficient to promote screw-like movement. The combination of flagellin expression I propose is the only one missing from Figure 1.

We agree with this Reviewer (and also Reviewer 1) and we have therefore constructed the corresponding FlaBA mutant and determined its swimming and spreading ability. The findings further strengthen our point and show that the FlaBA mutant does not form screws, and that the wild-type flagellin configuration is, in fact, advantageous. The results of the analysis of the FlaBA mutant have been added to Figure 1 and 2, as well as the appropriate controls to the corresponding supplemental figures. We also performed corresponding simulations, which have been added as Figure S10 d. The text has been rephrased to also accommodate these findings.

Lined 55. Slot “favorite” for “favorable”. Bacteria don’t have favorites.

This has been changed accordingly.

The section starting line 167 is very interesting. I agree with the authors interpretation that the first flagellin in the tandem array is expressed before the second, FliA-dependent flagellin and the substrate specificity switch and this preloads the cell with the proximal subunit. But I think what the data also says is that the first flagellin secretion stops immediately after the substrate specificity switch has been flipped. Has it ever been demonstrated as clearly as this that once the switch to late class flagellar proteins is made, that early class flagellar proteins are no longer secreted? Else, why does elongation stop? I believe current models would say otherwise that the chaperones control preferential late class protein secretion and outcompete, but not abolish, the early class. Very strange. Moreover, these data suggest something fundamental about T3SS specificity. If I’m understanding the constructs correctly, the only thing that differs is the promoter?

How would the promoter control secretion specificity? Or does activation of late class gene expression somehow completely inhibit early class? In short, please provide some explanation for why stubs occur when either protein is expressed from the early promoter.

The Reviewer raises a very important and intriguing point here, which we have not covered sufficiently in the previous version of the manuscript. We have therefore added more experimental data to this section. Most importantly, introduction of a second flaA copy under control of the flaA promoter increases the length of the proximal FlaA-segment, but does not double the length. Secondly, when overproducing FlaA (along with FliS) from a plasmid, the cells grow very long flagellar filaments (see Fig. S7c), and we have already shown in the previous version of our manuscript that, when producing FlaA from the flaB promoter, the filaments are of almost normal length (Fig. 1c,g). Taken together, this suggests (as the Reviewer correctly points out) that expression from the flaA promoter is downregulated or shut down upon activation of the flaB promoter. We have carried out some mutant studies (such as FlaA production in a fliA mutant to see if FliA might shut down the flaA promoter, and others) but we could not elucidate this mechanism so far, so we did not include the data here. We have amended the Results and Discussion parts accordingly. Although we fully agree that this a very interesting question, we are still convinced that, based on all of our results, the concept of sequential production is by far the most likely mechanism to reach the spatial distribution of flagellin within the filament. We have also phrased the conclusions more carefully and included the putative shutdown mechanism, which, we feel, would merit a separate study.

Paragraph starting 211. This paragraph was difficult to read and understand. Perhaps this can be improved by including figure citations at appropriate places between line 211 and 235.

We have rewritten this section completely and hope that is clearer now.

Fig 2a-f. What is P on the X-axis? Percentage?

This reviewer presumably referred to the Y-axis. P refers to the probability of observing the corresponding behavior (a certain run length, a certain velocity, a certain turning angle). This has now been added to the caption of Fig. 2.

Fig 2c-d. Why do some cells swim 10-fold faster than others? Are the “slow swimming cells” non-motile and moving by Brownian motion? At first I thought the slow moving cells were the screw like movements but then the FlaA only strain still has a slow moving subpopulation and the authors conclude that it can’t move like a screw.

We thank the Reviewer for this comment, and we agree that the complexity of the data shown take some 'unpicking' to fully understand. In the original submission, we wanted to convey that the slow-moving fraction of the population is simply that: cells that spend some time swimming more slowly. This fraction includes reversing/flicking cells, and those that simply swim slower. These are distinct from the screw-formers, however, which display behaviour like that shown in Fig. 2h – prolonged periods of slow but consistent swimming. We do not see these periods of sustained slow swimming in the FlaA-only strain, though there is a subpopulation of cells that simply swim more slowly. This explanation is supported by the fact that the slow-swimming portion of the population is not substantially enhanced in the presence of ficoll.

We are also confident that these cells truly are motile. We determine this by measuring the mean-squared displacement of all the objects that we detect and plotting it as a function of time on a log scale. The graph of a swimmer's mean squared displacement then has a slope of 2 in this representation, while the diffusing (Brownian-only) particles have a slope of 1, allowing for an objective discrimination between non-motile and motile particles, even if the swimming is slow. The non-swimmers (particles moving purely under Brownian motion) are excluded from our analysis. We have updated the supplementary methods section and added a specific comment in the main manuscript (materials and methods, 'holographic tracking') to state:

"For more details of the data reduction and analysis procedure, including the rejection of non-motile particles (those moving under Brownian motion only), please see the Supplementary Information."

Line 199. Clarify. Is this sentence referring to panel n?

We agree that this was not sufficiently clear. We have added the references to the corresponding Figure panels and hope that this is improved now.

Line 228. Omit "as expected" as it is not clear why this was expected.

Line 233. Clarify "this effect" as I'm not sure what effect is being described. Could this entire paragraph be rewritten for clarity? I've reread it three times and I don't have a clear idea of what the phenotype is. For instance, line 229 seems to indicate that FlaA-only cells are like wild type but line 231 seems to indicate that FlaA-only cells are like FlaB-only cells.

Paragraph 236. I don't understand this paragraph either. It says that single flagellin filaments have significant effects on swimming behavior and then in the next sentence says that FlaA-only mutants have no behavioral phenotype?

To all three points above: We thank this Reviewer for pointing out that this whole section was apparently not clear. We have rewritten this section as requested and hope that this has sufficiently improved the clarity.

Line 245. "The pronounced increase in slow swimming cells for the wild type" seems a strange way to frame the data. Shouldn't this be "the pronounced decrease in slow swimming cells for the FlaA-only strain" as the comparison is how the mutant differs from the wild type? Moreover, I don't understand the conclusions of this section. The biophysical analysis in figure 2d indicates that FlaA-only cells perform better in high viscosity and I think the model is that the screw behavior is advantageous at high viscosity but Fig 3a indicates that FlaA only cells are unable to conduct screw behavior. If the plate assay motility simulates microviscosity environments (dead ends) and the defect of the FlaA-only cells (Fig 1n) is the inability to adopt a screw, why then does FlaA-only overperform in high viscosity (Fig 2d)?

We thank this reviewer for highlighting the need for clarification here, as this is one of the major findings of our work. Cells with the FlaA-only filament perform quite well under free-swimming conditions, and it may be expected that, due to the increased stability, they also perform well (or better) in soft-agar plates in complex environments, although movement to soft agar represent a very different setting (line 226 ff). However, we found that under these conditions, the more unstable filaments (wild type and FlaB-only) best support spreading. We have previously proposed that screw formation is not advantageous for

free-swimming cells but gives an advantage when cells get stuck and the flagellar filament (wrapped around the cell body) comes into contact with the surface (as shown in Kühn et al., 2017). Since the FlaA-only filament cannot form the screw, FlaA-only cells are more likely to get stuck for good in soft agar. We therefore conclude that screw formation is a highly important feature for polarly flagellated cells to move in structured environments. We have elaborated on this in the discussion section (line 404 ff).

There needs to be an explicit statement as to how motility plates, viscosity, slow and fast swimming subpopulation and torque are related to each other. Otherwise, the paper reads somewhat like a shell game where phenotypes are revealed but you can't tell whether it was in any way related to a previously revealed result.

While rephrasing the paragraph, we have included more explanation on how the free-swimming parameters may relate to moving in structured environments, such as soft agar. We hope that this has improved the clarity. However, as stated in the answer to the previous question, swimming motility and spreading through soft agar are very different processes, and, as we found, more effects do play a role.

Line 258. "with wrapped-up flagella" is this jargon that is synonymous with "screw-like motion"?

This just described the behavior of the flagellar filament.

Line 259. "This swimming phenotype", clarify, there are many parameters being discussed.

Has been defined ('the screw-like swimming phenotype; line 255).

Line 258, Line 262. "Not a single" if not a single cell was observed, how is there a standard deviation in the graphs? Moreover, why is the comparison of two "not a single" result being clarified as "n.s." not significantly different in Fig 3a? By definition, identical results are not significantly different.

The error bars in Fig. 3a are in fact not standard deviations but 95 % confidence intervals (CI) that were calculated with the exact binominal test. Although some strains show "not a single" screw formation the data sets are not exactly identical because different total events were counted. Therefore, calculation of the CI and P-values with Fisher's exact test give slightly different results. However, the reviewer is absolutely right: If no screw formation was observed it doesn't make much sense to calculate significance at all, so we have excluded all the strains that don't show screw formation from these calculations. To avoid confusion we also removed the CI error bars for these strains, as they don't provide much helpful information.

Paragraph starting line 265. I have no idea what is going on in this paragraph, what figure 3b is, or how it relates to the data. I believe it to be a mathematical model that supports the need for FlaB to cause the flagellum to wrap around the body of the cell but the text supporting it was very hard to understand. Paragraph starting 280. I'm not really sure what this paragraph is about but I think it is more mathematical modeling?

Line 292. Mathematical simulations don't confirm anything. At best they support the biological observations.

We have rewritten that passage and added more explanation to the figure Legends. The simulation was not meant to 'prove' or 'confirm' the data, but to provide a possible explanation for our findings. The passages have also been toned down accordingly. We hope that it is clearer now what this simulation is about and which potential mechanism it suggested: just decreasing the diameter of the helix may be sufficient to stabilize the filament against wrapping and screw formation.

Is the 20% FlaA composition supported by the protein data where FlaA and FlaB can be distinguished by size in an SDS PAGE gel? FigS2a shows different ratios by immunoblotting which could be complicated by glycosylation altering antigen access. I believe there is also a coomassie stain located below each immunoblot (not explained in the figure legend). In the coomassie stain of panel A, there are bands that could be equal intensity?

As we use whole cell extracts, this is rather hard to predict. However, as the labeling is so very clear, we are convinced that the flagellin ratio within the filament is well reflected. Labeling of the immunoblots and SDS-PAGE has been added.

Reviewer #3 (Remarks to the Author):

This manuscript describes the role of multiple flagellins in the motility of *Shewanella*. Ignoring the presence or absence of a sheath, flagellated bacteria can (roughly) be placed into two categories. There are the enteric bacteria exemplified by *E. coli* & *Salmonella*. They express multiple flagella composed of a single flagellin type. True, most *Salmonella* have antigenically distinct flagellins, but they are functionally identical. Historically, these two bacteria have served as model systems for flagellar function, assembly, coupled gene regulation and chemotaxis. However, there is a whole other world of bacteria that have flagella composed of different flagellin types and the top of the regulatory hierarchy is completely different than the enteric FlhDC-control type. *Shewanella* is an excellent example of non-enteric bacteria that produce multiple flagellin types because it has only two subunit types making it easier to characterize than those with many more.

The authors first demonstrate that the unexpectedly large masses of FlaA and FlaB on gels is due to glycosylation and that glycosylation is lost in deletions of *S. oneidensis* flagellin modification orthologs *pseG* or *maf-1*. The authors need to explain why deletion either gene gives the same result. Are the sequential steps in a known glycosylation pathway?

Flagellin glycosylation in Shewanella (as well as in most other bacteria with similarly decorated flagellins) is still widely unknown terrain. Earlier studies by our and other groups on closely related S. oneidensis MR-1 showed that deletion of almost any of the genes encoding proteins involved in flagellin glycosylation results in the absence of the modification, and we assumed (and did see) the same for S. putrefaciens. We have added this to the results part (line 126).

The authors then demonstrate the spatial arrangement of FlaA and FlaB with FlaA polymerized at the filament base. It is clear that cells transition from FlaA to FlaB since *flaB* deletions produce short FlaA-filaments suggesting a mechanism in place that limits FlaA secretion. The authors then go on to show that FlaA (and FliS) are produced early in the regulatory cascade before FliA-dependent FlaB production, which nicely allows coordination of gene expression with FlaA-first assembly. Also, expressing *flaA* from the *flaB* locus and vice versa demonstrated that the mechanism controlling short FlaA filament length was at the level of gene expression. Nice!

The authors go on to get at the reason for different flagellins by measuring numerous parameters - velocity, run durations, turning angles - under different environmental conditions - normal medium versus high viscosity (10% Ficoll). The increase in slow-swimming cells for the wild-type and FlaB-only mutant led the authors to test the effect of high load on filament instability and the screw-like slow mobility. This makes sense for organisms that have exposure to changing environmental conditions - liquid versus solid surface contact, sediments, etc., unlike enterics that live in a more homogeneous viscous environment. Thus, this represents a relatively comprehensive study, which provides a solid evidence

for their hypothesis as to the need for multiple flagellin types requires for a lifestyle in the real (non-enteric) world. Great paper!

We thank this Reviewer for the very positive and encouraging remarks.

Minor comments:

line 55: change "favorite" to "optimal" - how do you know what their "favorite niche is? Their true favorite niche might be a bar in Berlin.

As we did not want to speculate on the favorite niche of our particular species, we have changed this accordingly.

line 59: add "subunits" after "flagellin"

Has been added.

line 61: change "therefore" to "their expression is"

Has been changed.

line 118: "several"??? three or four??? I need a more accurate number here

We have counted one to six lateral flagella, which are produced by a subpopulation of S. putrefaciens. However, we have omitted this sentence due to space restrictions, as the lateral system does not play a role in this study and has been removed for the species studied here.

REVIEWERS' COMMENTS:

Reviewer #1 (Remarks to the Author):

In the revised version, the authors have addressed most of my concerns about the scientific comment, but I am still unsatisfied with clarity of the presentation. It is only by reading the paper alongside their response to reviewers that I felt comfortable with their complete argument, and I think much of what was communicated in the response should be better explained in the actual manuscript before it can be published. I do think the results are quite interesting so would like to see them communicated as well as possible.

Specifically,

1) The discussion of the sequential production model is split into two areas: the results, and the second paragraph of the discussion.

I believe that much of the material in the discussion should be moved to the results to help the reader understand a framework in which to understand the results. In particular, I think the material from "We propose (line 333) to "important length determining factor" (line 341) should all be in the results section, as they define the hypothetical model being tested and include results and interpretation of results that test that hypothesis.

A brief mention in the results earlier that the current data does not completely explain the stumps and cessation of FlaA production would also clarify the status of what is shown by the results, with more detailed discussion left to the discussion section.

2) At line 186, I believe FlaA-only refers to normal-length filaments constructed by putting flaA under control of the flaB promoter as in the previous paragraph. However, it is also possible that this could refer to the strain called FlaA-only on line 145, which is potentially confusing and should be clarified.

3) I agree with Reviewer 2 that the discussion of phenotypes on p7 is confusing and this still needs more clarity. To my reading, what makes this confusing is that there is both a distinction between "slow/fast swimming" as well as "distributed turning angles/only high turning angles", and as explained in the response, the slow swimmers include both flicking/"slow" swimmers as well as screw-type swimmers. The categories also intersect each other, i.e., some of the high reversal angle cases (FlaB-only) are screw swimmers, while others (FlaA-only) are not -- at least that is my understanding of the results reported on line 221.

First, I think the discussion of different types of slow swimmers (and how screw swimmers are a distinct type within that) needs to be in the main manuscript as well as the response letter.

Second, I think the distinction between the high reversal angles for FlaA-only and FlaB-only should be made explicit. Perhaps a summary paragraph before the numerical simulation section with the complete taxonomy of phenotypes and how they correspond to different filament composition and proposed flagellar conformation (i.e. extended vs screw-type) would help here.

4) Finally in the last month we published some results on *Helicobacter suis* in Scientific Reports that may be relevant to this paper.

Henry Fu

Reviewer #2 (Remarks to the Author):

The manuscript is wide in that it tries to do many things but each without sufficient depth to support the overarching claims. Unfortunately, the revision has reinforced my suspicion that the thesis is less than the sum of its potentially interesting parts. Each part does something but how the next builds from or informs the preceding is never clear.

The first section repeats a result previously shown for *S. oneidensis* in the closely related *S. putrefaciens*: that the flagellins are glycosylated. I do not know why this result is important or how it relates to the main conclusion. The authors claim this information could be used to determine the relative levels of the two proteins, but this could have been done without the glycosylation data (FlaA and FlaB are different sizes) and the relative levels aren't reported in this section. In short, this section could be omitted without any effect on the manuscript. Or rather, reduced to a single sentence and moved to the modeling section.

The second section is the most interesting to me. Here the authors use cell biology to show that the flagellins are spatially enriched in filament in manner dependent on the promoter that expresses them. Note, the spatial heterogeneity has been reported before and thus the novelty seems to be how transcription controls the order. This is a very unusual result and potentially very interesting but the mechanistic analysis is shallow and unsatisfying. They engineer strains to build full length filaments with only one type of flagellin and show that homopolymeric filaments do not expand well in soft agar chemotaxis assays.

The third section analyzes a number of properties of swimming behavior of wild type and the homopolymeric filament strains to try to account for the soft agar colonization defect. A number of parameters are carefully measured but I believe that the conclusion is ultimately, that the different flagellar compositions don't really have much of an effect on swimming to account for the agar colonization defect. Thus the analysis doesn't help answer the question from the previous section.

The fourth section focuses on screw-like motility, the ability of the flagellum to wrap around the body of *Shewanella* and rotate the entire body, somewhat like a spirochete. Their evidence suggests that FlaA is needed to be enriched at the base to oppose spontaneous adoption of the screw morphology. This is a nice observation but what it has to do with colonization in the soft agar assay is unexplained. Note, the ability to form or not form a screw seems independent of the soft agar defect as both FlaA-only and FlaB-only filaments both have a defect. Moreover, bacteria that cannot form screws, like *E. coli*, colonize soft agar well so the connection between soft agar colonization and screw formation seems spurious. Thus the analysis doesn't help answer the question from section 2.

The fifth section is mathematical modeling which concludes with the statement that the model suggests something about how the composition relates to behavior. I believe that the data is what makes the suggestion and that the model is just supporting the

observation with math.

So for me the paper boils down to figure 2 (and related supplemental) and figure 3a. I believe that the mixed filament composition resists screw formation, and that's an interesting correlation. But considered more broadly, I'm not certain that cells maintain multiple flagellins for this purpose (to control screw formation or for navigating complex environments) as the authors claim. *E. coli* functions in constrained environments, navigating pores in soft agar, without screw formation and screws seem to form in organisms that may not have mixed filaments (*Pseudomonas* and *Burkholderia* are mentioned in the discussion as forming screws but not in the introduction as examples of multiple flagellins). The evolutionary and functional correlation doesn't seem to hold.

I can't help but feel that the manuscript as a whole is an amalgam of multiple incomplete stories strung together and ultimately, I wasn't convinced that the paper showed what the title claimed "Spatial arrangement of several flagellins within bacterial flagella improves motility in different environments".

Authors' response to the reviewers' comments.

First of all, we would like to thank the Referees again for their efforts to improve the quality of our story and the corresponding manuscript. Please find below a point-by-point list in which we address the remaining issues.

REVIEWERS' COMMENTS:

Reviewer #1:

In the revised version, the authors have addressed most of my concerns about the scientific comment, but I am still unsatisfied with clarity of the presentation. It is only by reading the paper alongside their response to reviewers that I felt comfortable with their complete argument, and I think much of what was communicated in the response should be better explained in the actual manuscript before it can be published. I do think the results are quite interesting so would like to see them communicated as well as possible.

Dear Dr. Fu,

We are very grateful for the very positive and encouraging remarks and the efforts you have put into improving our story. Following your and the Editor's advice, we have added more explanations to the results section, e.g. short paragraphs summarizing the key findings, which also help to transition to the following Results sections. We agree that this improves the clarity of our findings significantly.

Specifically,

1) The discussion of the sequential production model is split into two areas: the results, and the second paragraph of the discussion.

I believe that much of the material in the discussion should be moved to the results to help the reader understand a framework in which to understand the results. In particular, I think the material from "We propose (line 333) to "important length determining factor" (line 341) should all be in the results section, as they define the hypothetical model being tested and include results and interpretation of results that test that hypothesis.

A brief mention in the results earlier that the current data does not completely explain the stumps and cessation of FlaA production would also clarify the status of what is shown by the results, with more detailed discussion left to the discussion section.

This is a very good suggestion. Following this, we have now moved the mentioned section of the discussion to the results part, and have added a sentence about the likely cessation of flaA promoter activity (line 174-182). We agree that this improves the clarity of this section. This passage reads:

Based on these results we propose that the sequential production pattern depends on two individual promoters driving the expression of flaA and flaB, respectively. The length of the proximal filament segments formed under control of the weaker FliA-independent flaA promoter showed a wide variation, indicating flagellin production from this promoter varies substantially at the single cell level. Accordingly, overexpression of FlaA from a plasmid resulted in aberrantly long flagellar filaments (Supplementary Fig. 7b, c), indicating that the amount of available flagellin monomers is an important length determining factor. However, the data did not explain why flagellin production from the flaA

promoter only produces filament stubs, which indicates cessation of the expression from this promoter that may occur at the onset of flaB expression.

Further explanations (partly moved up from the Discussion section) have been added to the numerical simulations (lines 325-329; 336-345).

2) At line 186, I believe FlaA-only refers to normal-length filaments constructed by putting flaA under control of the flaB promoter as in the previous paragraph. However, it is also possible that this could refer to the strain called FlaA-only on line 145, which is potentially confusing and should be clarified.

We fully agree that there is potential for confusion. We have therefore added the corresponding description, and the passage now reads: “We therefore compared the spreading ability of the wild-type cells (FlaAB) and FlaA-only (with *flaA* being expressed from the *flaB* promoter, resulting in filaments of normal length), FlaB-only and FlaBA mutants through soft agar.”
We also renamed the strain in line 145 to FlaA stub. We hope that this has become clearer now.

3) I agree with Reviewer 2 that the discussion of phenotypes on p7 is confusing and this still needs more clarity. To my reading, what makes this confusing is that there is both a distinction between "slow/fast swimming" as well as "distributed turning angles/only high turning angles", and as explained in the response, the slow swimmers include both flicking/"slow" swimmers as well as screw-type swimmers. The categories also intersect each other, i.e., some of the high reversal angle cases (FlaB-only) are screw swimmers, while others (FlaA-only) are not -- at least that is my understanding of the results reported on line 221.

First, I think the discussion of different types of slow swimmers (and how screw swimmers are a distinct type within that) needs to be in the main manuscript as well as the response letter.

Second, I think the distinction between the high reversal angles for FlaA-only and FlaB-only should be made explicit. Perhaps a summary paragraph before the numerical simulation section with the complete taxonomy of phenotypes and how they correspond to different filament composition and proposed flagellar conformation (i.e. extended vs screw-type) would help here.

We are grateful for pointing out the need for more clarification. According to the suggestion, we have now added a summarizing paragraph before the numerical simulation section in order to put the analysis of swimming and flagellar filament stability/ability to screw formation into context. This section recapitulates the connection between the increase of the slow-moving populations and screw formation as well as the advantage of screwing motility in soft agar. However, we do not see any obvious connection between screw formation and differences in turning angles, as both FlaA-only (no screws) and FlaB-only (many screws) mutants display a similar tendency for forward-backward movements. This paragraph now reads:

The findings of the screw formation analysis correlate well with the observed differences in free-swimming speed distributions and spreading through obstructed environments such as soft agar. Wild-type and FlaB-only filaments readily form flagellar screws under conditions of high viscosity, which increases the population of slow free-swimming cells (cp. Fig. 3c, d) as free swimming in screwing mode does not allow effective propulsion²³. Accordingly, this increase in the slow population of free-swimming cells is absent in FlaBA and FlaA-only mutants, which lack the ability of screw formation. However, the latter two strains are hampered in spreading through structured and obstructed environments, the soft agar plates (Fig. 1k, l), strongly indicating that screw formation is significantly contributing to movement when the cells can make contact with a solid environment. In contrast, the differences in efficient turning angles observed under free-swimming conditions at high viscosity cannot be attributed to screw formation: Compared to the wild type FlaB-only cells, that are prone to screw formation, as well as FlaA-only cells, which are incapable of screw formation, tend to turn at high angles in less efficient forward-backward movement. Notably, in contrast to the mutants, the wild-type cells driven by the native FlaAB filament always performed very well under all conditions tested.

4) Finally in the last month we published some results on *Helicobacter suis* in Scientific Reports that may be relevant to this paper.

We have noticed this paper, which was particularly interesting as H. suis is bipolarly flagellated and is still able to form the flagellar screw at one or both poles. We have now included this work in our discussion (line 416).

Reviewer #2:

The manuscript is wide in that it tries to do many things but each without sufficient depth to support the overarching claims. Unfortunately, the revision has reinforced my suspicion that the thesis is less than the sum of its potentially interesting parts. Each part does something but how the next builds from or informs the preceding is never clear.

We regret that this Reviewer was not convinced by the overall structure and flow of our story. To improve this, we have, also in response to Reviewer 1, Dr. Fu, tried to smoothen the transitions between the different parts. This will be elaborated below.

The first section repeats a result previously shown for *S. oneidensis* in the closely related *S. putrefaciens*: that the flagellins are glycosylated. I do not know why this result is important or how it relates to the main conclusion. The authors claim this information could be used to determine the relative levels of the two proteins, but this could have been done without the glycosylation data (FlaA and FlaB are different sizes) and the relative levels aren't reported in this section. In short, this section could be omitted without any effect on the manuscript. Or rather, reduced to a single sentence and moved to the modeling section.

The difference in the molecular masses of the two flagellins was important for the up-following studies on regulation later on, as it allowed us to determine if a single, both, or none of the flagellins was produced. We have also added this as transition to the end of this section (line 123-125).

The reviewer is right that it was maybe not absolutely required to show that the mass difference is due to differences in glycosylation, but we felt that this should better be demonstrated rather than just speculated. Notably, without glycosylation, the two flagellins are indistinguishable by PAGE/immunoblotting as they are almost the same size (28.6 vs. 28.4 kDa; p4, 20) and not different as the Reviewer implies here. Therefore, and as asked for by the Editor, we would like to keep this section, but we have shortened it.

The second section is the most interesting to me. Here the authors use cell biology to show that the flagellins are spatially enriched in filament in manner dependent on the promoter that expresses them. Note, the spatial heterogeneity has been reported before and thus the novelty seems to be how transcription controls the order. This is a very unusual result and potentially very interesting but the mechanistic analysis is shallow and unsatisfying. They engineer strains to build full length filaments with only one type of flagellin and show that homopolymeric filaments do not expand well in soft agar chemotaxis assays.

While we fully agree that some open questions remain (e.g., with respect to the apparent cessation of expression from the flaA promoter), we are convinced that our set of experiments was sufficient to a) show that the spatial flagellin arrangement very likely relies on sequential production based on separate regulation of flaA and flaB. More importantly, we were b) able to produce mutants with different filament compositions, which enabled us to perform the key experiments that show how the composition affects swimming, spreading, and stability of the

filament, which is the major part of this study. To make the general findings clear, we have added a paragraph summarizing the main and important results (p6, 1-9). This paragraph reads:

Based on these results we propose that the sequential production pattern depends on two individual promoters driving the expression of flaA and flaB, respectively. The length of the proximal filament segments formed under control of the weaker FliA-independent flaA promoter showed a wide variation, indicating flagellin production from this promoter varies substantially at the single cell level. Accordingly, overexpression of FlaA from a plasmid resulted in aberrantly long flagellar filaments (Supplementary Fig. 7b, c), indicating that the amount of available flagellin monomers is an important length determining factor. However, the data did not explain why flagellin production from the flaA promoter only produces filament stubs, which indicates cessation of the expression from this promoter that may occur at the onset of flaB expression.

The third section analyzes a number of properties of swimming behavior of wild type and the homopolymeric filament strains to try to account for the soft agar colonization defect. A number of parameters are carefully measured but I believe that the conclusion is ultimately, that the different flagellar compositions don't really have much of an effect on swimming to account for the agar colonization defect. Thus, the analysis doesn't help answer the question from the previous section.

This reviewer nearly perfectly summarizes the most important point of this section: There are significant differences in free-swimming capability, but these do NOT explain the differences observed for spreading in soft agar. This was meant to be elaborated in the summarizing paragraph of this section, which also highlighted the differences between free swimming and spreading through structured environments. We have therefore rephrased this paragraph. In addition, we think a possible misunderstanding may arise from the conception that a structured environment can be simulated by increasing the viscosity – which is not the case. To make this clearer, this fact is now explicitly stated in this summary and transition section (line 174-182).

The fourth section focuses on screw-like motility, the ability of the flagellum to wrap around the body of Shewanella and rotate the entire body, somewhat like a spirochete. Their evidence suggests that FlaA is needed to be enriched at the base to oppose spontaneous adoption of the screw morphology. This is a nice observation but what it has to do with colonization in the soft agar assay is unexplained. Note, the ability to form or not form a screw seems independent of the soft agar defect as both FlaA-only and FlaB-only filaments both have a defect.

Regarding the final statement: we suspect that there was a general misunderstanding, as our data very clearly shows a correlation between screw formation and spreading through soft agar: the two strains able to form flagellar screws (the wild type and the FlaB-only mutant) spread very well, while the FlaA-only and FlaBA mutant had a significant spreading defect. To make this clearer we have, also following the suggestion of Reviewer 1, Dr. Fu, added a paragraph summarizing the main findings of swimming, spreading and screw formation to the end of the fourth section (line 283-296) before the transition into the simulation section.

The findings of the screw formation analysis correlate well with the observed differences in free-swimming speed distributions and spreading through obstructed environments such as soft agar. Wild-type and FlaB-only filaments readily form flagellar screws under conditions of high viscosity, which increases the population of slow free-swimming cells (cp. Fig. 3c, d) as free swimming in screwing mode does not allow effective propulsion²³. Accordingly, this increase in the slow population of free-swimming cells is absent in FlaBA and FlaA-only mutants, which lack the ability of screw formation. However, the latter two strains are hampered in spreading through structured and obstructed environments, the soft agar plates (Fig. 1k, l), strongly indicating that screw formation is significantly contributing to movement when the cells can make contact with a solid environment. In contrast, the differences in efficient turning angles observed under free-swimming conditions at high viscosity cannot be attributed to screw formation: Compared to the wild type FlaB-only cells, that are prone to

screw formation, as well as FlaA-only cells, which are incapable of screw formation, tend to turn at high angles in less efficient forward-backward movement. Notably, in contrast to the mutants, the wild-type cells driven by the native FlaAB filament always performed very well under all conditions tested.

Moreover, bacteria that cannot form screws, like *E. coli*, colonize soft agar well so the connection between soft agar colonization and screw formation seems spurious. Thus the analysis doesn't help answer the question from section 2.

We have never claimed in this manuscript that screw formation is crucial for spreading, but we are convinced that we can definitely show here for *S. putrefaciens* that the ability of screwing motility gives a significant advantage for moving through structured environments, and this is likely true for more bacterial species with polar (!) flagellar systems. Spreading of species with peritrichous flagellation patterns, such as *E. coli*, should not be directly compared with that of polarly flagellated species, as they likely have developed different mechanisms to cope with structured environments.

The fifth section is mathematical modeling which concludes with the statement that the model suggests something about how the composition relates to behavior. I believe that the data is what makes the suggestion and that the model is just supporting the observation with math.

The model is supporting the observation, and, in addition, shows that the mechanical properties can already be explained by the different geometry of the filaments consisting of FlaA or FlaB, which is a very nice addition to the experimental data. We have added this as a passage to the first paragraph of this section (line 325-329), which reads:

Thus, the simulation not only reproduces our experimental findings but also suggests that introducing a proximal FlaA filament segment with a smaller diameter and pitch is already sufficient to result in the observed stabilization of the flagellar filament against screw formation, indicating the geometry of the helix as a key factor in the determination of the mechanical properties.

We also summarized the major conclusions at the end of this section (line 336-345):

*The data indicate a variation of the onset of screw formation depending on the FlaA fraction and show that in a range between 16 and 20% the cells achieve efficient forward propulsion and maintain the ability to form screws at high torque. This is well within the range of proximal segments we determined for wild-type cells (~8% to ~28%; Fig. 4b). With respect to this pronounced variability in length of the proximal FlaA segment within the flagellar filament, the experimental data in concert with the simulations strongly suggest that *S. putrefaciens* forms a highly heterogenous cell population with respect to propulsion and the ability of screw formation. This may not be optimal at the single cell level but distributes the trade-off between torque generation and propulsive efficiency among the population, which allows efficient spreading of at least a fraction of the population under a wide range of conditions.*

We hope that the contributions of the simulations have become clearer now.

So for me the paper boils down to figure 2 (and related supplemental) and figure 3a. I believe that the mixed filament composition resists screw formation, and that's an interesting correlation. But considered more broadly, I'm not certain that cells maintain multiple flagellins for this purpose (to control screw formation or for navigating complex environments) as the authors claim. *E. coli* functions in constrained environments, navigating pores in soft agar, without screw formation and screws seem to form in organisms that may not have mixed filaments (*Pseudomonas* and *Burkholderia* are mentioned in the discussion as forming screws but not in the introduction as examples of multiple flagellins). The evolutionary and functional correlation doesn't seem to hold.

I can't help but feel that the manuscript as a whole is an amalgam of multiple incomplete stories strung

together and ultimately, I wasn't convinced that the paper showed what the title claimed "Spatial arrangement of several flagellins within bacterial flagella improves motility in different environments".

As already mentioned above, we have never claimed that multiple flagellins are a requirement for screw formation and efficient spreading. However, we are convinced that we have demonstrated here (for our model species) that the spatial organization of different flagellins provides a means to create a flagellar filament best suited for moving through a range of different environments. This likely is one, but probably not the only reason for the maintenance of multiple flagellins within bacteria, and we expect it to be similarly be true for a number of (but probably not all) bacterial species with multiple flagellins.